# The genetic organization of longitudinal subcortical volumetric change is stable throughout the lifespan

Anders Martin Fjell[1,2]*, Hakon Grydeland[1], Yunpeng Wang[1], Inge K Amlien[1], David Bartres-Faz[3], Andreas M Brandmaier[4,5], Sandra Düzel[4], Jeremy Elman[6], Carol E Franz[6], Asta K Håberg[7,8], Tim C Kietzmann[9,10], Rogier Andrew Kievit[9], William S Kremen[6,11], Stine K Krogsrud[1], Simone Kühn[4,12], Ulman Lindenberger[4,5], Didac Macía[3], Athanasia Monika Mowinckel[1], Lars Nyberg[1,13], Matthew S Panizzon[6,14], Cristina Solé-Padullés[3], Øystein Sørensen[1], Rene Westerhausen[1], Kristine Beate Walhovd[1,2]

[1]Center for Lifespan Changes in Brain and Cognition, Department of Psychology, University of Oslo, Oslo, Norway; [2]Department of Radiology and Nuclear Medicine, Oslo University Hospital, Oslo, Norway; [3]Departament de Medicina, Facultat de Medicina i Ciències de la Salut, Universitat de Barcelona, and Institut de Neurociències, Universitat de Barcelona, Barcelona, Spain; [4]Center for Lifespan Psychology, Max Planck Institute for Human Development, Berlin, Germany; [5]Max Planck UCL Centre for Computational Psychiatry and Ageing Research, Berlin, Germany; [6]Center for Behavioral Genomics Twin Research Laboratory, University of California, San Diego, La Jolla, United States; [7]Department of Radiology and Nuclear Medicine, St. Olavs Hospital, Trondheim University Hospital, Trondheim, Norway; [8]Department of Neuromedicine and Movement Science, Faculty of Medicine and Health Sciences, Norwegian University of Science and Technology (NTNU), Trondheim, Norway; [9]MRC Cognition and Brain Sciences Unit, University of Cambridge, Cambridge, United Kingdom; [10]Donders Institute for Brain, Cognition and Behaviour, Radboud University, Nijmegen, Netherlands; [11]Center of Excellence for Stress and Mental Health, VA San Diego Healthcare System, La Jolla, United States; [12]Lise Meitner Group for Environmental Neuroscience, Max Planck Institute for Human Development, Berlin, Germany; [13]Department of Radiation Sciences, Umeå Center for Functional Brain Imaging, Umeå University, Umeå, Sweden; [14]Department of Psychiatry, University of California, San Diego, La Jolla, United States

*For correspondence:
andersmf@psykologi.uio.no

Competing interests: The authors declare that no competing interests exist.

**Abstract** Development and aging of the cerebral cortex show similar topographic organization and are governed by the same genes. It is unclear whether the same is true for subcortical regions, which follow fundamentally different ontogenetic and phylogenetic principles. We tested the hypothesis that genetically governed neurodevelopmental processes can be traced throughout life by assessing to which degree brain regions that develop together continue to change together through life. Analyzing over 6000 longitudinal MRIs of the brain, we used graph theory to identify five clusters of coordinated development, indexed as patterns of correlated volumetric change in brain structures. The clusters tended to follow placement along the cranial axis in embryonic brain development, suggesting continuity from prenatal stages, and correlated with cognition. Across independent longitudinal datasets, we demonstrated that developmental clusters were conserved through life. Twin-based genetic correlations revealed distinct sets of genes governing change in

each cluster. Single-nucleotide polymorphisms-based analyses of 38,127 cross-sectional MRIs showed a similar pattern of genetic volume–volume correlations. In conclusion, coordination of subcortical change adheres to fundamental principles of lifespan continuity and genetic organization.

## Introduction

Cortical development follows a topographic organization through childhood and adolescence (*Fjell et al., 2019*; *Krongold et al., 2017*; *Raznahan et al., 2011*). This means that regions of the cortex with similar structural and functional properties tend to develop together (see *Eickhoff et al., 2018* for a discussion of cortical topography in the context of neuroimaging). This topography is conserved through later development and aging (*Fjell et al., 2019*; *Tamnes et al., 2013*) and follows the genetic organization of the cortex, i.e. is controlled by overlapping sets of genes (*Fjell et al., 2015*). It is not known whether the same is true for subcortical structures. In contrast to the monotone thinning of the cerebral cortex (*Storsve et al., 2014*), lifespan trajectories of subcortical structures are more diverse and complex (*Allen et al., 2005*; *Narvacan et al., 2017*; *Raznahan et al., 2014*; *Walhovd et al., 2005*). This may be due to fundamental ontogenetic and phylogenetic differences between cortical and subcortical regions. The embryonic origin of the cortex is the pallium, while cerebellar and subcortical structures originate from the hindbrain, diencephalon, or subpallium (*Tuller et al., 2008*). These structures can be placed according to their position along the cranial vertical axis (see Table 2). Although the subcortex is evolutionary older than the cortex, it has a higher proportion of evolutionarily more recent genes, and a higher evolutionary rate, which is a basic measure of evolution at the molecular level (*Tuller et al., 2008*). It has also been argued that genes expressed in the subcortex generally are more region specific (*Tuller et al., 2008*; *Zhang and Li, 2004*). These mechanisms may be seen in human development and aging, with higher plasticity and potential for change in response to environmental impacts in phylogenetically older structures (*Walhovd et al., 2016b*), especially the hippocampus (*Engvig et al., 2014*; *Eriksson et al., 1998*). This combination of plasticity and vulnerability could contribute to the larger diversity in the lifespan trajectories of subcortical structures (*Walhovd et al., 2005*). On the other hand, a hypothesis is that genetically governed neurodevelopmental processes can be traced in the brain later in life (*Chen et al., 2011*; *Satizabal et al., 2019*). This would for instance entail that brain regions under shared genetic control in development continue to be influenced by the same genes and change together through life. This has been shown for the comparably less plastic cortex (*Fjell et al., 2015*). In light of the diverse age trajectories and high plasticity of subcortical structures, it is not known whether patterns of subcortical maturation in childhood can be traced back to principles of embryonic development, how developmental organization sets constraints on subcortical aging, and the degree to which this organization of change is under common genetic control. The aim of the present study was to address these unresolved issues about the organization of subcortical change across the lifespan. Specifically, we tested how subcortical developmental volumetric change clustered across different structures, how similar this organization was in development versus aging, and whether clusters of change were influenced by shared genetics. We hypothesized that volumetric changes in the developmental structures would tend to cluster according to embryonic principles, i.e. placement along the cranial vertical axis, that the pattern of change in aging would be similar to the pattern of change detected in childhood, and that structures changing together throughout the life would be governed by the same sets of genes.

## Results

### Clusters of change in development

First, we determined which regions showed correlated change in development. A single-center longitudinal dataset (Center for Lifespan Changes in Brain and Cognition [LCBC]), comprising 974 healthy participants from 4.1 to 88.5 years with a total of 1633 MRI examinations, was used. The sample was divided into development and adulthood/aging (development [<20 years], n = 644, 1021 MRIs, follow-up interval = 1.7 years [1.0–3.2]; adulthood/aging [≥20 years], n = 330, 612 MRIs, follow-up interval = 1.6 years [0.2–6.6]), see *Fjell et al., 2015* for details (sample descriptives are

provided in *Table 1*). Annual symmetrized percent change (APC) was calculated for each participant for each brain region, averaged over hemispheres, using the formula APC = (Vol Tp2 – Vol Tp1) / (Vol Tp2 + Vol Tp1) × 100. If more than two time points were available, the first and the last were used to calculate APC. These APCs were correlated across participants between each pair of brain regions. The Louvain algorithm for detecting communities in networks (*Blondel et al., 2008a*) was applied to derive clusters in the correlation matrix from the development sample, and the Mantel test run to compare the different matrices. Five clusters of coordinated developmental change were identified (*Figure 1*) (see *Cluster stability analyses* and *Validation analyses* below for a more detailed discussion and justification of the cluster solution). The optimized community-structure statistic Q, the so-called modularity, ranges between −1 and 1, and measures the relative density of connections within communities as compared to links between communities (*Girvan and Newman, 2002*). We compared the cluster solution's Q value with the Q from 10,000 randomized networks preserving the signed degree distribution and rewiring each edge approximately five times using the *randmio_und_signed* function in the BCT. The community-structure solution was significantly more clustered than in the random networks (p<0.001, developmental change Q=0.44, the 2.5 and 97.5 percentile of the random Q distribution=0.36–0.40). Three large clusters consisted of the ventricles (Cluster 1); the brain stem, cerebellum white matter (WM) and cerebellum cortex, cortical WM, thalamus and hippocampus (Cluster 2); and cortex, putamen, amygdala, and nucleus accumbens (Cluster 3). Caudate (Cluster 4) and pallidum (Cluster 5) were represented by separate clusters.

## Pattern of subcortical change in adulthood and aging can be predicted from development

Next, we wanted to test whether the topographical organization of change detected in development was conserved in adulthood and aging. To this end, the pairwise change–change correlations between regions were calculated for the adult and aging sample (*Figure 2*), and the Mantel test was run to compare the developmental and the adult/aging matrices. The change–change matrices were more similar than expected by chance, r = 0.72 (p<0.0001), demonstrating substantial overlap of clusters of change in development and aging. The results were replicated using longitudinal data from the Lifebrain consortium (total n = 756, 1512 MRIs, mean follow-up interval = 2.3 years, age 19–89 years, mean 59.8 years), yielding almost identical results (r = 0.71, p<0.0001).

## Cluster stability analyses

We tested the stability of the identified developmental clusters. As different clustering approaches often yield different results, we ran a series of post hoc analyses to confirm the validity of the cluster solution. In our main analysis, we decomposed the correlation matrices into clusters or modules, where each module comprised regions that were more densely connected to each other – based on its correlation value – and sparsely connected to regions in other modules, by means of the commonly employed Louvain modularity algorithm (*Blondel et al., 2008a*). We proceeded to assess the stability and validity of the identified clusters using alternative approaches. As a sensitivity analysis, we performed consensus clustering (*Lancichinetti and Fortunato, 2012*; *Romero-Garcia et al., 2018*) combined with the *versatility* metric to aid selection of the resolution parameter γ for the Louvain algorithm (*Shinn et al., 2017*). In the main analysis, we used the default resolution parameter γ

**Table 1.** Sample overview.

| Sample | N | N longitudinal | Observations | Age Mean (range) | Sex Female/Male | Interval years Mean (range) |
|---|---|---|---|---|---|---|
| LCBC | 974 | 635 | 1633 | 25.8 (4.1–88.5) | 508/466 | 2.3 (0.2–6.6) |
| VETSA* | 331 | 331 | 662 | 56.3 (2.6) | 0/331 | 5.5 (0.5) |
| Lifebrain[x] | 756 | 756 | 1512 | 59.0 (19.3–89.0) | 330/426 | 2.2 (0.3–4.6) |
| UKB | 38,127 | na | 38,127 | 63.6 (44–81) | 20,026/18,101 | na |
| UKB long | 1337 | 1337 | 2674 | 62.5 (46–80) | 663/674 | 2.3 (2–3) |

*75 complete monozygote (MZ)/53 complete dizygote (DZ) pairs of male twins.
[x]Not including LCBC.

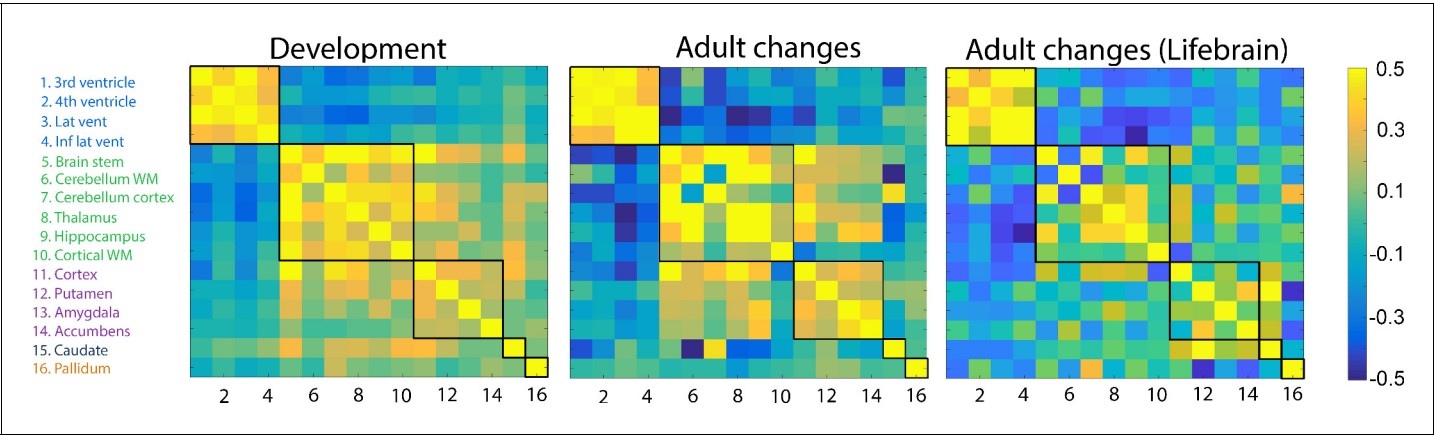

**Figure 1.** Volumetric change–change relationships. Heatmaps represent pairwise correlation coefficients between volume change (annualized percent change) of the brain structures in development in the LCBC sample (left), aging in the LCBC sample (middle), and aging in the Lifebrain replication sample (right). The five clusters, delineated by the black lines, were derived from the developmental sample.

= 1, and using this for the consensus clustering yielded an identical solution. As the $\gamma$ value increases from 0, the decomposition yields a progressively larger number of modules. Versatility provides an objective function to guide the choice of the resolution parameters. Specifically, the mean versatility across regions depends on how consistently the region affiliate with a specific module. An 'optimal' value of $\gamma$ is therefore defined as a value for which the versatility is lowest, i.e. for which the decomposition is the least ambiguously defined. The mean versatility does not provide an objective global optimum of the resolution parameter $\gamma$; instead, it serves to guide optimization of $\gamma$ to local minima within neighborhoods corresponding to the desired spatial resolution of the modules. Here, we calculated mean versatility across a range of resolution values, by re-running the Louvain algorithm (via the *find_optimal_gamma_curve* function from *Shinn et al., 2017* and the consensus function therein) across the resolution range $0.01 \leq \gamma \leq 4.00$, with increments of 0.01, and 1000 runs per $\gamma$ value. Seven local minima of mean versatility were identified ($\gamma$ = 0.66, 0.84, 0.93, 1.03, 1.12, 1.17, and 1.45). The final community partition at each $\gamma$ was defined as a consensus across another 1000 runs of the Louvain modularity algorithm at the selected value of the resolution parameter. The results showed that beyond the local minima of $\gamma \leq 0.64$, yielding $\leq$ three clusters, and $\gamma \geq 1.91$ yielding 50% singleton modules ($\geq$ eight clusters), the $\gamma$ of 1.17–1.21 had the lowest versatility (versatility = 0). These $\gamma$ levels all yielded the five-cluster solution of the main analysis. Also, $\gamma$ = 0.7 and y = 0.96 yielded the same five-cluster solution. $\gamma$ = 1.32 and $\gamma$ = 1.35, the latter which also had a very low

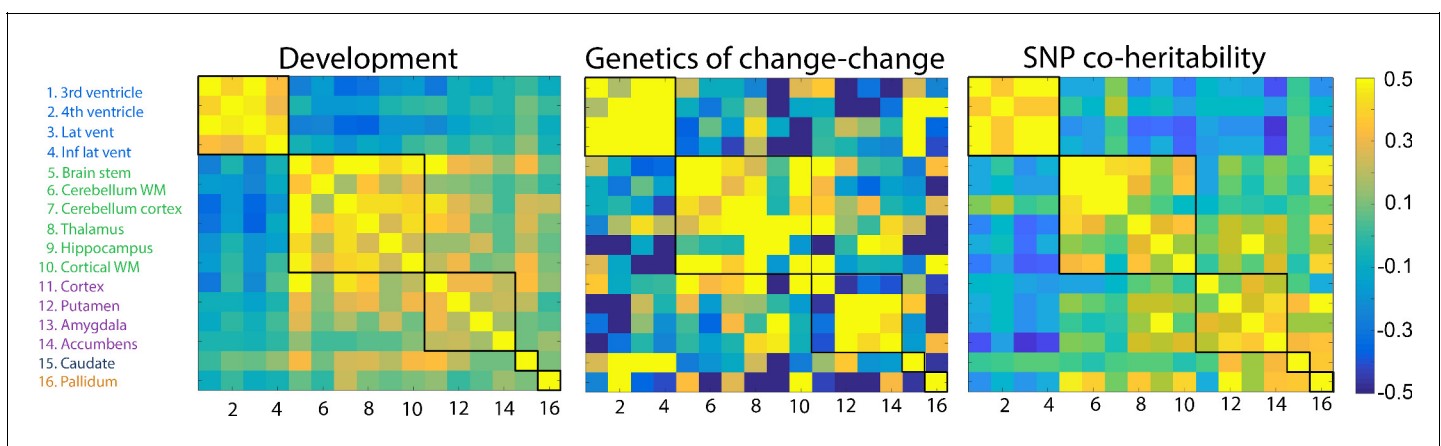

**Figure 2.** Genetic correlations. Left: Change–change correlations in development used to generate clusters. Middle: Genetic change–change correlations, i.e.the genetic contribution to the relationships between change among any two structures, based on twin analysis (VETSA). Right: SNP genetic correlations from the UKB cross-sectional data. The five clusters, delineated by the black lines, were derived from the developmental sample.

versatility of 0.001, yielded a six-cluster solution. γ = 1.60 and γ = 1.78 yielded a seven-cluster solution. Thus, the five-cluster solution was the most frequent and showed the lowest versatility over several γ levels. Specifically, the five-cluster solution appeared seven times compared with once for the three- and eight-cluster solutions and twice for the six- and seven-cluster solutions. This supported the stability of the initial solution. In the three-cluster solution, one cluster consisted of the ventricles and one consisted of pallidum, amygdala, and accumbens, while the remaining structures were included in the last cluster. Finally, we tested the similarities of the community structure (cluster solution) of the developmental and the adult/aging change–change matrices (*Betzel and Bassett, 2017*). We calculated the normalized mutual information (NMI) (*Lancichinetti and Fortunato, 2009*), variation of information (VI) (*Meilă, 2003*), and the z-score of the Rand coefficient (*Red et al., 2011*) using the Network Community Toolbox (http://commdetect.weebly.com). Null distributions were constructed by comparing the community structure of the developmental change–change matrix with the community structure derived from 10,000 randomized networks preserving the signed degree distribution of the adult/aging matrix. All of these metrics supported the conclusion that the developmental and the adulthood/ aging change matrices were more similar than expected by chance (NMI = 0.65, p=0.003; VI = 0.96, p=0.0002; zRand = 5.33, p<0.0001). Due to the nature of the research questions and data, which included gray matter (GM) and WM compartments as single structures, the null models generated were not spatially constrained (*Alexander-Bloch et al., 2018*; *Burt et al., 2020*). This may have increased the similarities between change matrices and partitions.

## Patterns of change adhere to principles of genetic organization

An important part of the study was to assess whether regions within the same clusters showed shared genetic influences. Using multivariate latent change score models, we calculated the differences in subcortical volumes from baseline to follow-up in the Vietnam Era Twin Study of Aging (n = 331, mean follow-up interval = 5.5 years) and computed pairwise genetic correlations of the slope factor (i.e., change) between all regions (see *Brouwer et al., 2017* for details on the statistical twin model). This yielded an estimate of how much of the change–change relationship between any two brain structures is due to common genetic influence. The latent change score model accounts for the relatedness between the twins. As the model is genetically informative, the relationship between the twins is fundamentally built into the model. The latent change score model was preferred over the simpler difference score model for the twin analyses, as it utilizes multiple sources of information within time to estimate change, restricts measurement error to the level of the observed variables, and allows for the estimation of covariance with intercept. The result is possibly a more precise estimation of change over the assessment window.

The matrix of shared genetic influences on change between each pair of brain structures (*Figure 2*) was tested against the developmental and the adult/aging change–change correlation matrices. The Mantel test confirmed that the shared genetic influences on the change–change relationships were statistically more similar to the pattern of correlated changes during development (r = 0.46, p=9.999e$^{-05}$) and aging (r = 0.37, p<0.0002) than expected by chance. Replication was again run using Lifebrain data, yielding r = 0.37 (p<0.0004) between the matrix of shared genetic influences on change and the Lifebrain aging change–change correlation matrix.

In order to further explore the genetic contributions to coordinated subcortical change, we first attempted to calculate the pairwise single-nucleotide polymorphism (SNP)-based genetic correlation between change in each pair of structures by running a mega-analysis on 1337 participants with longitudinal MRIs from UK Biobank and 508 from LCBC. However, this initial analysis showed that statistical power – as could be expected – was too low to yield valid estimates. Thus, we instead based the SNP genetic analyses on the cross-sectional UKB data where power is much greater (n = 38,127, age 40–69 years), using age, sex, and the first 10 components of the genetic ancestry factor as covariates. A detailed overview of the pairwise genetic correlations is presented in *Supplementary file 1* – SNP-based heritability estimates (Legend: pairwise co-heritability between brain structures derived from 38,127 participants from UKB). The Mantel test (r = 0.57, p<0.0005) demonstrated that the SNP genetic correlation matrix was more similar to the developmental change matrix than expected by chance. This showed that the genetic organization of subcortical structures in middle-age can be predicted from the organization of change during brain development in childhood. For completeness, we also compared the SNP genetic correlation matrix to the aging change matrix (r = 0.56, p<0.0003) and the heritability of coordinated change matrix from the VETSA sample (r = 0.38,

p<0.0003), in both cases showing significantly higher similarity than expected by chance. The UKB SNP genetic correlation matrix also showed excellent coherence with the Lifebrain replication sample (r = 0.47, p<0.0006) and with the change–change matrix derived from the 1337 UKB participants with longitudinal MRI (r = 0.46, p<0.0002).

## Embryonic origins of subcortical organization

We tested the hypothesis that development in childhood mimics the organization of earlier embryonic development. An overview of the clusters and their embryonic developmental origins is given in *Table 2*. Although a one-to-one correspondence between embryonic development and clustering of change in childhood was not expected, there were clear tendencies to conservation of embryonic developmental principles in later childhood development. The regions of Cluster 2 mainly emerged from rhombencephalon or the posterior prosencephalon, making up structures placed low on the cranial vertical axis, including brain stem (myelencephalon), cerebellum cortex and WM (metencephalon), and the thalamus (diencephalon). The exception to this was that the hippocampus and the cortical WM were also included in Cluster 2. The extensive connectivity between cerebellum and cerebrum, and the similarities in development of WM in cerebellum and cerebrum, may explain the latter finding. Clusters 3–5 comprised structures developed from subpallium/ventral telencephalon (caudate, pallidum, putamen) and pallium/dorsal encephalon (amygdala, cortex), also showing consistency with the major principles from embryonic development and placement along the cranial vertical axis.

Directly to explore the relationship between principles of embryonic development and adult genetics, we applied the Louvain algorithm on the UKB SNP genetic correlation matrix. This allowed us to detail how shared genetic influences were distributed across structures in the cross-sectional UKB data. The results were then mapped according to the main stages of early brain development, from the primary brain vesicles through the secondary brain vesicles and to the developed structures. A two cluster solution yielded a trivial divide between a ventricular cluster and one cluster containing the remaining structures. Thus, we ran a separate analysis on the non-ventricular structures. This revealed a match between the adult genetic clusters and their embryonic origins (*Figure 3*). Pallidum, putamen, nucleus accumbens, and caudate clustered together, all originating from the subpallium (ventral telencephalon), which is the developmental origin of the basal ganglia. Amygdala, hippocampus, and the cerebral cortex clustered together, having the pallium (dorsal telencephalon) as common embryonic origin. Brainstem, cerebellum WM, and cerebellum cortex, all from the

**Table 2.** The embryonic origins of the clusters and placement along the cranial vertical axis.

| Brain structure | Cluster | Embryonic development | | | Cranial vertical axis |
|---|---|---|---|---|---|
| Third ventricle | 1 | Prosencephalon (posterior) | Diencephalon | | |
| Fourth ventricle | 1 | Rhombencephalon | | | |
| Lat ventricle | 1 | Prosencephalon (anterior) | Telencephalon | | |
| Inf lateral ventricle | 1 | | | | |
| Brainstem (medulla oblongata) | 2 | Rhombencephalon | Myelencephalon | | 0 |
| Cerebellum cortex | 2 | Rhombencephalon | Metencephalon | | 1 |
| Cerebellum WM | 2 | Rhombencephalon | Metencephalon | | 1 |
| Thalamus | 2 | Prosencephalon (posterior) | Diencephalon | | 2 |
| Hippocampus | 2 | Prosencephalon (anterior) | Telencephalon (dorsal) | Pallium (medial) | 4 |
| Cortical WM | 2 | Prosencephalon (anterior) | Forebrain WM | | |
| Caudate | 4 | Prosencephalon (anterior) | Telencephalon (ventral) | Subpallium | 3 |
| Pallidum | 5 | Prosencephalon (anterior) | Telencephalon (ventral) | Subpallium | 3 |
| Putamen | 3 | Prosencephalon (anterior) | Telencephalon (ventral) | Subpallium | 3 |
| Accumbens | 3 | Prosencephalon (anterior) | Telencephalon (ventral) | Subpallium | 3 |
| Amygdala | 3 | Prosencephalon (anterior) | Telencephalon (dorsal) | Pallium (lateral) | 4 |
| Cortex | 3 | Prosencephalon (anterior) | Telencephalon (dorsal) | Pallium (dorsal) | 4 |

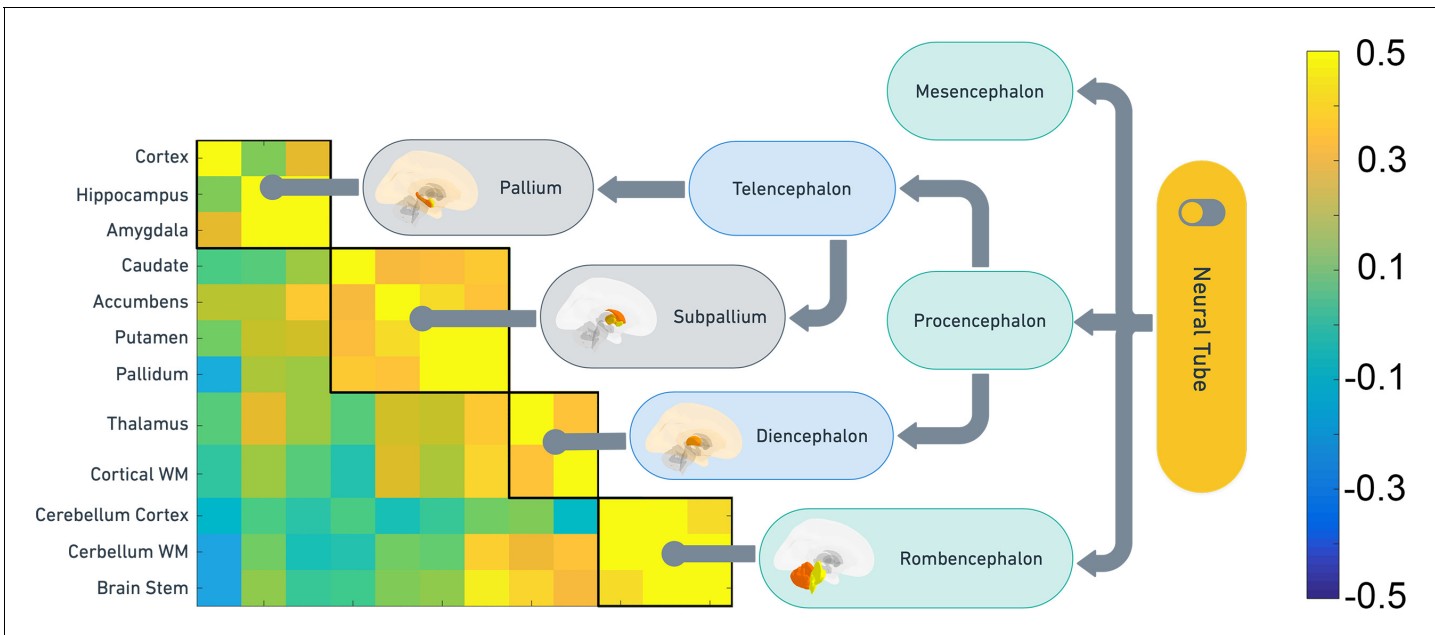

**Figure 3.** Correspondence between SNP heritability and embryonic brain development. Clustering of the non-ventricular structures was used to test how shared genetic variance were organized in the UKB sample, and the clusters were compared to the main organization of embryonic brain development. The heatmap shows the pairwise genetic correlations. The flow chart shows the main features of embryonic brain development and how the genetic clusters obtained from middle-aged and older adults follow the same organization.

rhombencephalon, formed a third cluster, while thalamus (diencephalon) and the cerebral WM constituted the last. Although this analysis is auxiliary to the main change-analyses reported above, it demonstrates a common fundamental principle of close correspondence between embryonic brain development and the brain's genetic architecture decades later.

## Age trajectories

Each cluster and regional volume was expected to yield unique age trajectories during development and adulthood. To detail these, we fitted the developmental trajectory of each cluster by using the total volume of the structures within each cluster, with sex and intracranial volume (ICV) as covariates, by generalized additive mixed models (GAMM) (*Wood, 2006*; *Figure 4*, *Table 3*). Both Akaike information criterion (AIC) and Bayesian information criterion (BIC) were calculated to select among models and guard against over-fitting. These analyses were done to assess differences in the trajectories between clusters. Since the total volume was used, large structures would potentially influence the cluster trajectories more than would smaller structures. Cluster 1 increased linearly, although the rate of increase was modest. Cluster 2 showed a decreasing exponential function with volume increase leveling off after 15 years. Cluster 3 mimicked a cubic relationship, with a slight increase in volume until about 8 years, then steeper reductions, which were gradually smaller from 15 years. Cluster 4 (caudate) showed an inverted U-shaped trajectory with a sharp increase until about 9 years, and Cluster 5 (pallidum) showed a cubic trajectory with similarities to Cluster 3. Next, we calculated aging-trajectories for each of the clusters defined in the developmental sample and. The trajectories across the adult age-range differed qualitatively between clusters in terms of steepness and shape. The trajectory for each cluster represented a continuation of the developmental trend seen in childhood and adolescence. Cluster 1 showed an exponential increase, Cluster 2 an inverted U-shaped trajectory, Clusters 3 and 4 almost linear reductions, while Cluster 5 showed reductions until about 50 years and little or no change after that. For completeness, the age trajectories of the clusters were also fitted across the full age-range from 4.1 to 88.5 years (numeric results in SI).

Finally, we also estimated the age trajectories of each volume of 16 brain regions (*Figure 5*, *Table 4*).

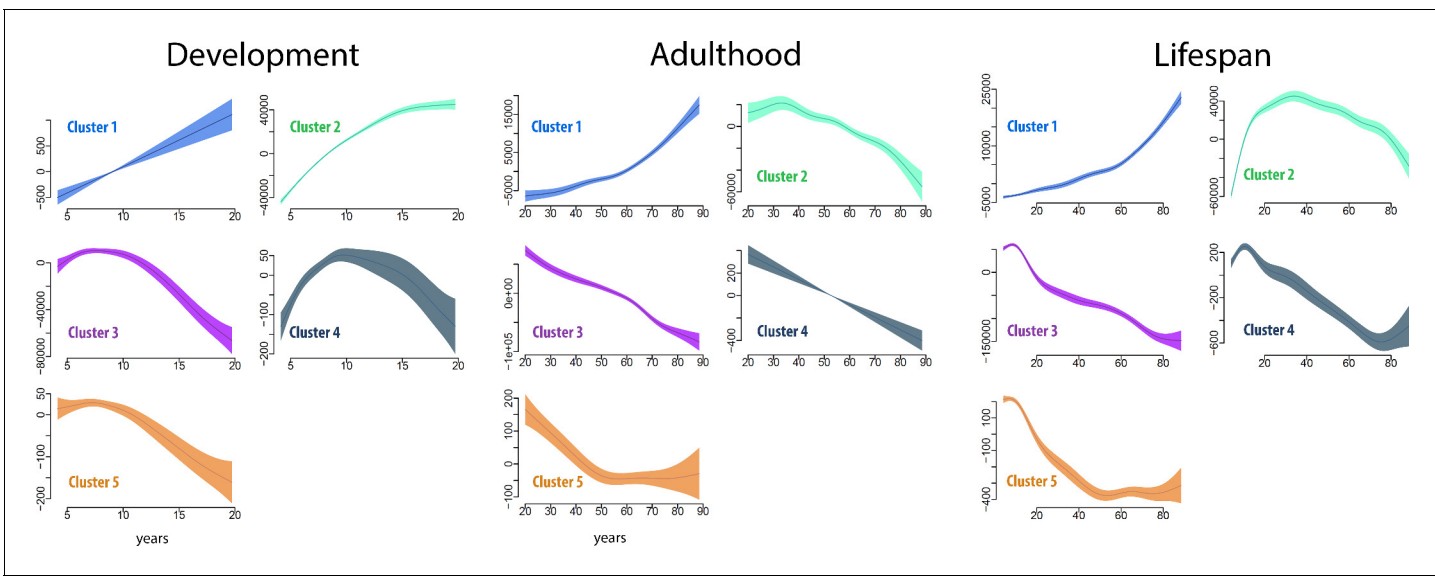

**Figure 4.** Cluster age trajectories for each cluster, for development (left), adulthood (middle), and the full lifespan (right). The trajectories are fitted with GAMM, and the shaded areas represent 95% CI. Note that the y-axes scales vary for easier viewing. The trajectories were estimated for development and adulthood separately to ensure that the analyses were fully independent.

Auxiliary analyses were done relating the clusters to general cognitive function (GCA) as measured by the Wechsler's Abbreviated Scale of Intelligence (*Wechsler, 1999*) in the full LCBC sample, using sex and age as covariates. Corrected for five comparisons, two clusters showed significant (p<0.01) relationships to GCA, that is Cluster 1 (t = 3.74, p=0.00018) and Cluster 3 (t = 2.75, p=0.006). Low GCA score was associated with lower volumes. The relationships survived including intracranial volume (ICV) as covariate. ICV was not controlled for in the initial analyses since it is expected to follow brain volume development in the first part of life. For Cluster 1 was a significant interaction between GCA and age found (F = 4.59, p=0.01), while for Cluster 3, age trajectories did not differ significantly as a function of GCA (all p's>0.46), thus showing stable relationships across life.

## Validation analyses

A cluster solution will ultimately depend on which brain regions that are included and how different parameters for the clustering algorithm are defined. To test the validity of using the clusters defined in development across the adult and genetic samples, we ran two-sample Student's t-tests to assess whether the mean intra-cluster correlation was larger than the mean extra-cluster correlation. The

**Table 3.** Cluster age trajectories.

Numeric results for the trajectory analyses in *Figure 4*. Edf: effective degrees of freedom (signifying the complexity of the trajectory, where the value two approximates a quadratic shape, 3 a cubic shape, etc). The p-value is associated with the null hypothesis that there is no relationship to age.

| | Development | | | Adulthood and aging | | | Lifespan | | |
|---|---|---|---|---|---|---|---|---|---|
| | Edf | F | p | Edf | F | p | Edf | F | p |
| Cluster 1 | 1.1 | 51.0 | $0.23e^{-12}$ | 6.0 | 67.1 | $2e^{-16}$ | 7.5 | 176.6 | $2e^{-16}$ |
| Cluster 2 | 6.5 | 363.5 | $2e^{-16}$ | 6.8 | 16.0 | $2e^{-16}$ | 8.7 | 214.3 | $2e^{-16}$ |
| Cluster 3 | 5.4 | 37.4 | $2e^{-16}$ | 6.7 | 85.5 | $2e^{-16}$ | 8.6 | 206.7 | $2e^{-16}$ |
| Cluster 4 | 5.7 | 18.1 | $2e^{-16}$ | 1.0 | 79.4 | $2e^{-16}$ | 8.3 | 45.9 | $2e^{-16}$ |
| Cluster 5 | 3.7 | 16.9 | $3.33e^{-12}$ | 3.8 | 15.8 | $1.23e^{-11}$ | 7.9 | 171.0 | $2e^{-16}$ |

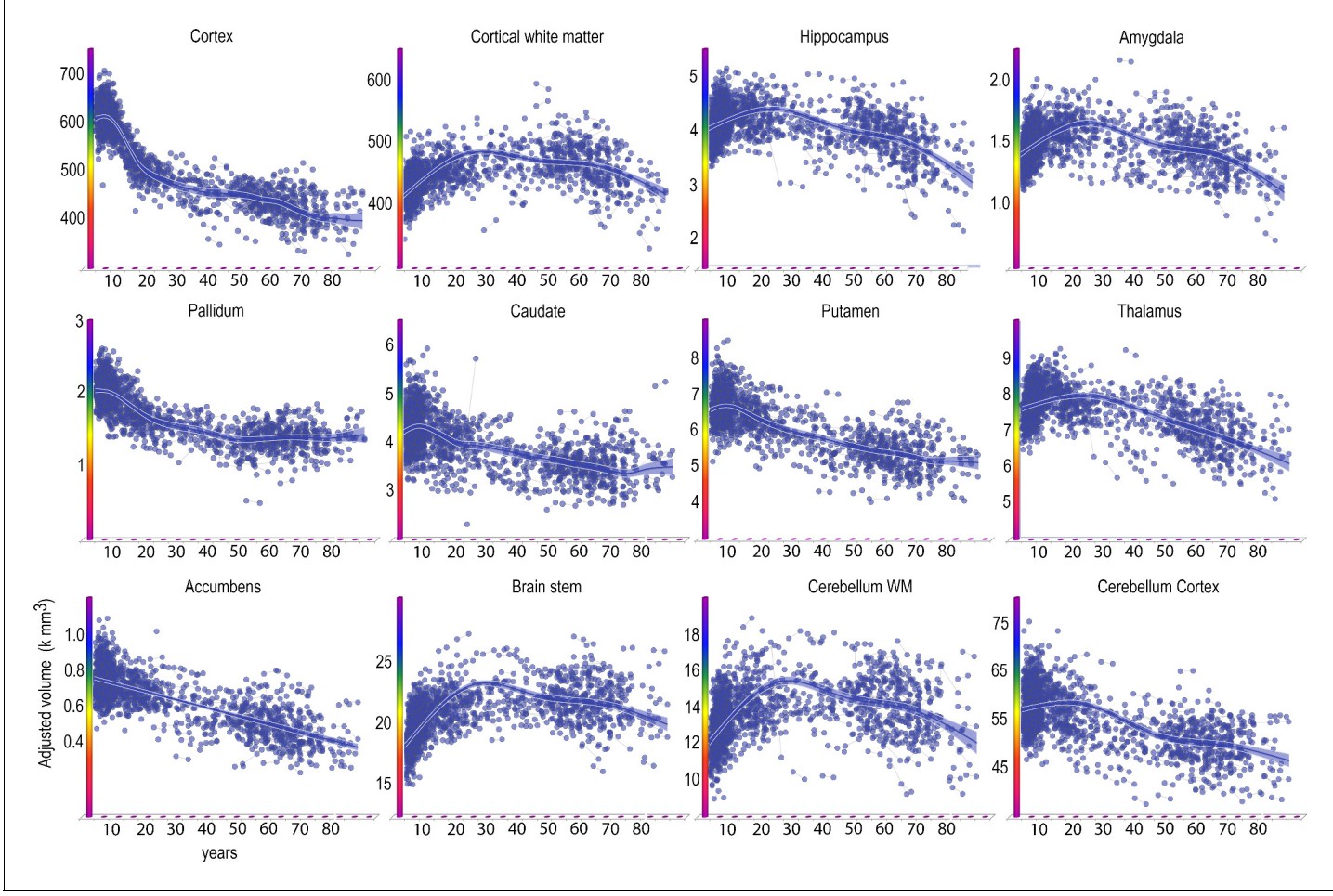

**Figure 5.** Lifespan trajectories of brain volumes. Age on the x-axis, volume in units of milliliters on the y-axis. The trajectories are fitted with GAMM, using both longitudinal and cross-sectional data, and the shaded areas represent 95% CI. Y-axis is in units of 1000 mm³. Ventricular volumes not shown.

intra-cluster correlations were significantly higher than the extra-cluster correlations in all the tested samples, except for Cluster 3 in the VETSA sample (*Table 5*). As the cluster solution was defined in an independent dataset, this outcome supports the validity of each cluster, and complements the Mantel test which is based on the full correlation matrix. Second, as changes in different parts of the ventricular system were expected to be highly correlated, we re-ran the Mantel tests excluding the ventricles. As expected, the coherence between matrices was reduced, but was still significant for all comparisons, except UKB (Dev vs LCBC adulthood and aging r = 0.44, p<0.009; Lifebrain r = 0.41, p<0.0008; VETSA r = 0.44, p<0.0009; UKB r = −0.06, p=0.63). This was in line with the higher within-cluster than between-cluster correlations reported above for the non-CSF(cerebrospinal fluid) clusters. Finally, we ran the Mantel test comparing the patterns of developmental to adult and aging changes in 12 genetically defined cortical regions from the same LCBC participants (see *Chen et al., 2013*; *Fjell et al., 2015*). This yielded a correlation of r = 0.83, which suggests that the organization of developmental subcortical change is conserved through life, but to a somewhat lesser extent than the organization of cortical change.

## Discussion

The results demonstrate that volumetric change of subcortical structures in development forms meaningful clusters. These clusters tend to follow the main cranial vertical axis from embryonic brain development, suggesting continuity from earlier to later stages of development. Although the lifespan trajectories of subcortical structures are more divergent than those for cortical regions

**Table 4.** Generalized additive mixed model fits LCBC lifespan.

Generalized additive mixed models (GAMM) were run with each neuroanatomical volume as dependent variable, and age, estimated total intracranial volume, and sex as covariates. Separate models were run with a linear age (age) term or a slope function (s(Age)). Except for cortex and caudate, the slope function yielded the lowest IC values. GM: Gray matter. WM: White matter. AIC: Akaike information criterion. BIC: Bayesian information criterion.

| | AIC | | BIC | | Effect of sex |
| --- | --- | --- | --- | --- | --- |
| | Age | S(Age) | Age | S(Age) | p |
| Accumbens | 18,324 | 18,335 | 18,357 | 18,367 | 0.57 |
| Amygdala | 20,360 | 20,048 | 20,392 | 20,081 | 0.13 |
| Brainstem | 27,514 | 26,805 | 27,546 | 26,837 | 0.11 |
| Caudate | 22,636 | 23,558 | 22,669 | 23,591 | 0.75 |
| Cerebellum cortex | 30,079 | 29,946 | 30,111 | 29,979 | 0.94 |
| Cerebellum WM | 27,430 | 27,017 | 27,463 | 27,050 | 0.16 |
| Cortex | 37,525 | 37,895 | 37,557 | 37,927 | 0.72 |
| Cortical WM | 36847 | 36,258 | 36,879 | 36,290 | 0.31 |
| Hippocampus | 22,494 | 22,235 | 22,526 | 22,268 | 0.22 |
| Pallidum | 20,937 | 20,741 | 20,969 | 20,774 | 0.17 |
| Thalamus | 23,841 | 23,669 | 23,874 | 23,701 | 0.06 |
| Total GM | 38,130 | 37,849 | 38,163 | 37,881 | 0.78 |
| Lateral ventricles | 30,276 | 30,147 | 30,308 | 30,174 | 0.33 |
| In flat vent | 21,081 | 20,931 | 21,113 | 20,964 | 0.23 |

(*Fjell et al., 2014*; *Walhovd et al., 2011*), the clusters were conserved through life. Thus, similar to what has been shown for the cerebral cortex (*Fjell et al., 2015*), regional subcortical volumetric changes in aging follow a similar pattern as developmental changes in childhood. This makes a strong case for the theory that early life sets the stage for aging (*Jagust, 2016*; *Walhovd et al., 2016a*; *Yeatman et al., 2014*). Importantly, the volumetric correlations within each cluster and the coordinated volumetric change of each cluster tended to be governed by common sets of genes. This conclusion was supported by the similarity of the volumetric change pattern during development with the pattern of genetic change–change correlations in the VETSA twin study and the cross-sectional UKB SNP co-heritability. Thus, coordination of volumetric change across subcortical structures adheres to similar principles of lifespan continuity and genetic organization previously seen for the cortex. This supports the hypothesis that genetically governed neurodevelopmental processes can be traced in subcortical structures throughout life.

**Table 5.** Within- vs. outside cluster correlations.

Two-sample Student's t-tests were run to test whether the mean correlation within the developmentally defined clusters ($r_i$) was larger than the mean correlation between the variables in the cluster and the variables outside the cluster ($r_e$). Note that the clusters were defined in the developmental sample, which is independent from the other four samples.

| | Cluster 1 | | | Cluster 2 | | | Cluster 3 | | |
| --- | --- | --- | --- | --- | --- | --- | --- | --- | --- |
| Dataset | $r_i$ | $r_e$ | p< | $r_i$ | $r_e$ | p< | $r_i$ | $r_e$ | p< |
| UKB cross-sectional | 0.52 | −0.10 | $1e^{-8}$ | 0.30 | 0.05 | $1e^{-7}$ | 0.28 | 0.06 | $9e^{-7}$ |
| VETSA (TWIN heritability) | 0.57 | −0.11 | $1e^{-8}$ | 0.40 | −0.05 | $3e^{-7}$ | 0.16 | −0.02 | 0.17 |
| Lifebrain (Aging) | 0.47 | −0.16 | $1e^{-8}$ | 0.14 | −0.05 | 0.0002 | 0.14 | −0.01 | 0.004 |
| LCBC (Aging) | 0.45 | −0.20 | $1e^{-8}$ | 0.30 | −0.06 | $1e^{-8}$ | 0.28 | 0.05 | $1e^{-7}$ |

$r_i$: intra-cluster correlation. $r_e$: extra-cluster correlation.

# Superordinate structures of change in childhood mimic embryonic brain development

The topographical development of cortical thickness is well described, with regional variability that follows certain functional, anatomical, and genetic principles of organization (*Chen et al., 2013*; *Fjell et al., 2019*; *Fjell et al., 2015*; *Raznahan et al., 2011*; *Tamnes et al., 2010*). Compared to the cerebral cortex, subcortical structures show highly divergent lifespan trajectories (*Fjell et al., 2013*; *Ostby et al., 2009*; *Raznahan et al., 2014*) and follow fundamentally different ontogenetic and phylogenetic principles (*Tuller et al., 2008*; *Zhang and Li, 2004*). On this background, the present results are intriguing. Meaningful clusters of developmental volumetric change were identified, with divergent developmental trajectories. Importantly, the different regions tended to cluster according to their embryonic origins. For instance, the majority of Cluster 2 structures emerged from rhombencephalon or the posterior prosencephalon, which is the origin of structures placed low on the cranial vertical axis. This included the brain stem (myelencephalon), cerebellum cortex, and cerebellum WM (metencephalon) and the thalamus (diencephalon). Clusters 3–5 comprise structures developed from subpallium/ ventral telencephalon (caudate, pallidum, putamen) and pallium/dorsal encephalon (amygdala, cortex). There were also departures from this principle, i.e. the inclusion of the hippocampus and the cortical WM in the same cluster. Hippocampus is part of the cerebral cortex, but develops from the medial pallium in contrast to the neocortex. Compared to many regions in the cerebral cortex, hippocampus has a more similar appearance across the range of mammal species (*Bingman and Salas, 2009*). Cortical WM was placed in the same cluster. Cortical WM originates from the anterior prosencephalon, but the major part of myelination occurs postnatally, so this compartment of the brain is not easily placed within the same embryonic developmental context. In addition, the WM label is anatomically gross with substantial regional differences in development (*Tamnes et al., 2010*). This cluster is also characterized by relatively high myelin content among several of its constituents, which may have contributed to the inclusion of hippocampus and cortical WM. Nevertheless, at a general level, change in the structures tended to cluster according to trends from embryonic development and placement along the cranial vertical axis. It must be noted as a limitation that the coherence between the developmental clusters and embryonic brain development is based on a qualitative judgment.

## Consistency in patterns of change across the lifespan

We found that subcortical structures that developed together during childhood tended to change together in adulthood and aging. Mapping the developmental clusters to the adult part of the sample yielded highly different change trajectories. Except for Clusters 3 and 4, which were characterized by mostly linear reductions, differences in the shapes of the slopes were observed. This suggests that clusters identified in development continued to show independent trajectories of change through the rest of life. The lifespan trajectories showed expected shapes, with accelerated increases for the ventricular system (Cluster 1) and an inverse U-shape for cluster 2, consisting of structures such as WM, hippocampus, and the brain stem (*Fjell et al., 2013*; *Walhovd et al., 2005*). Clusters 3, 4, and 5 showed different variants of initial increase in childhood, followed by relatively linear decline through most of adulthood that leveled off at high age, with variations between clusters in terms of break points. Clusters 4 (caudate) and 5 (pallidum) consisted of only one structure each. Cluster 3, however, consisted of four structures (cortex, putamen, amygdala, and nucleus accumbens), mimicking the known trajectory of cortical volume with more or less linear decline after the peak is reached in late childhood or early adolescence.

## Genetic organization of subcortical structure and change

We found that longitudinal volumetric change in regions that cluster together tend to show overlapping genetic influences. This conclusion was supported by the twin analysis, yielding higher genetic change–change correlations within versus than between clusters. This suggests that regions that develop and change together through life to a certain degree are influenced by shared sets of genes. Furthermore, the SNP analysis showed that also the cross-sectional volumetric relationships followed a similar organization. Thus, there seems to be a genetic basis for the consistent pattern of change in subcortical structures from development through the rest of life. It must be noted that although significant heritability estimates for brain changes have been demonstrated in ENIGMA,

evidence for genetic variants specific for brain change was found for cerebellar GM and the lateral ventricles only (*Brouwer et al., 2017*). Although limited statistical power prevented strong conclusions, genetic influence on volumetric change rates and baseline volume tended to overlap for most structures. Thus, we used cross-sectional data to further explore the genetic contribution to subcortical volumetric organization. Although these data do not reflect change, they increased sample size for these analyses from 6000 to more than 38,000 MRIs. Using the expanded sample for the SNP analyses, we found further evidence that regions within each developmental cluster tended to show overlapping genetic influences. A previous multi-sample GWAS also reported significant genetic correlations between some of the subcortical structures tested in the present study (*Satizabal et al., 2019*). The similarity of the developmental volumetric change–change matrix and the SNP genetic volume–volume correlation matrix from middle-aged adults thus yielded further support for the hypothesis that genetically governed neurodevelopmental processes can be traced through life. Moreover, the results revealed close correspondence between the genetic organization of subcortical structures and their embryonic origins, which suggest a link from embryonic brain development to the brain's genetic architecture in adulthood and aging. It must also be mentioned that high genetic correlations were identified between some structures from different clusters. This was especially true for hippocampus and amygdala, which showed high genetic change–change correlations in VETSA and genetic cross-sectional correlations in UKB.

### Limitations: Caveats in interpreting brain changes from MRIs and further research

Similar to all studies based on in vivo imaging methods, this study provides approximations of the underlying neurobiology. The MRI-derived measures are estimations, and the segmentations are based on signal intensities and contrast properties that are prone to the influence of multiple factors (*Walhovd et al., 2017*). Underlying mechanisms of volume differences and change are complex and may involve events such as aborization of axons and dendrites, axonal sprouting and loss, dendritic degeneration, vascular elaboration, synaptic pruning, as well as growth and reductions of myelination (see *Fjell and Walhovd, 2010* for a discussion of these issues). Many of these likely affect both contrast, signal intensity and volumetric estimations, but the relative effect of each is challenging to tease apart. We have previously shown age differences in cortical GM–WM contrast (*Westlye et al., 2009*) and T1w signal intensities (*Westlye et al., 2010*), which can also be detected longitudinally (*Vidal-Piñeiro et al., 2016*). The present results thus reflect effects of various neurobiological events on signal intensities and contrast. For instance, as mentioned above, myelin content has a major impact on T1w intensities, and myelin content is strongly related to age in development and aging (*Grydeland et al., 2019*). Thus, the clustering results will likely partly reflect different myelin content in the structures analyzed, as changes in myelin may be correlated across regions in the brain. A promising avenue for further research is to use multi-modal neuroimaging with different MRI sequences and analysis methods to yield more insight in the foundations for the volumetric changes. A second caveat is that although the clustering of regions is based on pairwise change–change correlations, this does not imply that each cluster consists of homogenous regions. Still, regions within a cluster showed more correlated volumetric change and higher genetic correlations with other regions within the cluster than with regions outside the cluster.

### Conclusion

Subcortical childhood development can be described according to meaningful clusters, which are stable through life, tend to follow gradients of embryonic brain development, and tend to be influenced by shared sets of genes. Thus, the pattern of change in subcortical regions may best be understood in a lifespan perspective.

## Materials and methods

### Samples

Multiple independent samples were used (*Table 1*). Details for all samples are found in SI.

## LCBC lifespan sample

A total of 1633 valid scans from 974 healthy participants (508 females/466 males), 4.1–88.5 years of age (mean visit age 25.8, SD 24.1), were drawn from studies coordinated by the Research Group for Lifespan Changes in Brain and Cognition (LCBC), Department of Psychology, University of Oslo, Norway (*Fjell et al., 2015*). For 635 participants, one follow-up scan was available, while 24 of these had two follow-ups. Mean follow-up interval was 2.30 years (0.15–6.63 years, SD 1.19). Sample density was higher in childhood/ adolescence than adulthood, since we expected more rapid changes during that age period (1006 observations < 10 years, 378 observations ≥ 20 and < 60 years, and 249 observations 60–88.5 years). All participants' scans were examined by a neuroradiologist and deemed free of significant injuries or conditions. The studies were approved by the Regional Committee for Medical and Health Research Ethics South, Norway (2010/2359; 2010/3407; 2009/200). Written informed consent was obtained from all participants older than 12 years of age and from a parent/guardian of volunteers under 16 years of age. Oral informed consent was obtained from all participants under 12 years of age.

## VETSA

Three hundred and thirty-one male twins (150 MZ/106 DZ paired twins/75 unpaired) were randomly recruited from the Vietnam Era Twin Registry and had imaging data at two time points. The study was approved by the Institutional Review Board at the University of California, San Diego. Written informed consent was obtained from all participants. Average age at baseline was 56.3 (2.6) years and follow-up interval 5.5 (0.5) years (see *Kremen et al., 2013*; *Kremen et al., 2006*). Based on demographic and health characteristics, the sample is representative of US men in their age range (*Kremen et al., 2013*; *Schoeneborn and Heyman, 2009*).

## The Lifebrain Consortium

Seven hundred and fifty-six participants with longitudinal MRI were included from the European Lifebrain project (1672 scans, baseline age 19–89 [mean = 59.8, SD = 16.4], mean follow-up interval 2.3 years, range 0.3–4.9, SD = 1.2) (https://www.lifebrain.uio.no/) (*Fjell et al., 2019*), including major European brain studies: Berlin Study of Aging-II (BASE-II) (*Bertram et al., 2014*; *Gerstorf et al., 2016*), the BETULA project (*Nyberg et al., 2020*), the Cambridge Centre for Ageing and Neuroscience study (Cam-CAN) (*Shafto et al., 2014*), and University of Barcelona brain studies (*Abellaneda-Pérez et al., 2019*; *Rajaram et al., 2016*; *Vidal-Piñeiro et al., 2014*). The study was approved by the Regional Committee for Medical and Health Research Ethics South, Norway (2017/653). Participants were screened to be cognitively healthy and in general not suffer from conditions known to affect brain function, such as dementia, major stroke, multiple sclerosis, etc. Exact screening criteria were not identical across sub-samples (see *Fjell et al., 2021* for details).

## UK Biobank

Thirty-eight thousand one hundred and twenty-seven participants with available MRIs and quality checked (QC) genetic information were included in the final analyses from UKB (40–69 years), see https://biobank.ndph.ox.ac.uk/. UKB has approval from the North West Multi-centre Research Ethics Committee (MREC). We received called genotypes for 488,377 subjects, of whom 40,055 had available MRIs pre-processed by FreeSurfer v6.0. We performed quality control of the genotype data at the participant level by removing participants failing genotyping QC (n = 550) or with abnormal heterozygosity values (n = 969). In addition, we removed 481 participants suggested to be removed for genetic analysis by the UK Biobank team. Ninety-one of these 481 participants had abnormal heterozygosity values, and the remaining were flagged out as outliers in heterozygosity/missing rate from the current QC files (ukb_sqc_VZ.csv) provided the most recent UK Biobank team. After excluding these subjects, we further remove related subjects by computing kinship coefficients using the program PLINK (*Chang et al., 2015*), with the option `-kinship` 0.0625. This amount to remove one subjects that are within the third degree of relatedness to any other participant. At variant level, we removed SNPs having minor allele frequency less than 0.01 or Hardy–Weinberg equilibrium test p-value < $10^{-6}$. In total, 784,356 SNPs were used in the subsequent analysis. We used the bivariate linear mixed model with genome-based restricted maximum likelihood methods implemented in the program GCTA (*Yang et al., 2011*) to compute genetic correlations for the volume measures for

each pair of the 16 brain subcortical structures. The principal components were computed using the above quality-controlled genotypes, after removing correlated SNPs with the option –indep-pairwise 100 50 0.1 from PLINK. In addition, 1331 participants with genotyping and longitudinal MRIs available were used as an additional replication sample to test stability of the phenotypic change–change pattern.

## Cognitive testing

GCA was assessed by WASI (*Wechsler, 1999*) for participants aged 6.5–89 years of age, while scores for corresponding subtests (Vocabulary, Similarities, Block design, and Matrices) from the Wechsler Preschool and Primary Scale of intelligence – III (WPPSI-III) (*Wechsler, 2008*) were used for the youngest participants (<6.5 years) (see *Walhovd et al., 2016a*). All participants scored within normal IQ range (82–145) or normal range of scaled scores (mean of subtests, s = 6.67–17.33).

## MRI data acquisition and analysis

Imaging data for the LCBC sample were acquired using a 12-channel head coil on a 1.5-Tesla Siemens Avanto scanner (Siemens Medical Solutions, Erlangen, Germany) at Oslo University Hospital Rikshospitalet and St. Olav's University Hospital in Trondheim (see *Walhovd et al., 2016a*). See *Table 6* for details regarding scanners and sequences.

For all samples, subcortical volumes were obtained by use of FreeSurfer (http://surfer.nmr.mgh. harvard.edu/) (*Dale et al., 1999*; *Dale and Sereno, 1993*; *Fischl et al., 2002*), processed with the longitudinal stream (*Reuter et al., 2012*). Specifically an unbiased within-subject template space and image (*Reuter and Fischl, 2011*) is created using robust, inverse consistent registration (*Reuter et al., 2010*). Several processing steps, such as skull stripping, Talairach transforms, atlas registration as well as spherical surface maps and parcellations are then initialized with common information from the within-subject template, significantly increasing reliability and statistical power (*Reuter et al., 2012*). For children, the issue of movement is especially important, as it could potentially induce bias in the analyses (*Reuter et al., 2015*). All children MRIs were manually rated for movement on a 1–4 scale, and only scans with ratings 1 and 2 (no visible or only very minor possible signs of movement) were included in the analyses, reducing the risk of movement affecting the

**Table 6.** Scanner and acquisition parameters.

| Sample | Scanner | Field strength (Tesla) | Sequence parameters |
|---|---|---|---|
| LCBC | Avanto Siemens | 1.5 | TR: 2400 ms, TE: 3.61 ms, TI: 1000 ms, flip angle: 8°, slice thickness: 1.2 mm, FoV: 240 × 240 m, 160 slices, iPat = 2 |
| | Avanto Siemens | 1.5 | TR: 2400 ms, TE = 3.79 ms, TI = 1000 ms, flip angle = 8, slice thickness: 1.2 mm, FoV: 240 × 240 mm, 160 slices |
| Barcelona | Tim Trio Siemens | 3.0 | TR: 2300 ms, TE: 2.98, TI: 900 ms, slice thickness 1 mm, flip angle: 9°, FoV: 256 × 256 mm, 240 slices |
| BASE-II | Tim Trio Siemens | 3.0 | TR: 2500 ms, TE: 4.77 ms, TI: 1100 ms, flip angle: 7°, slice thickness: 1.0 mm, FoV: 256 × 256 mm, 176 slices |
| Betula | Discovery GE | 3.0 | TR: 8.19 ms, TE: 3.2 ms, TI: 450 ms, flip angle: 12°, slice thickness: 1 mm, FoV: 250 × 250 mm, 180 slices |
| Cam-CAN | Tim Trio Siemens | 3.0 | TR: 2250 ms, TE: 2.98 ms, TI: 900 ms, flip angle: 9°, slice thickness 1 mm, FoV: 256 × 240 mm, 192 slices |
| UKB | Skyra Siemens | 3.0 | TR: 2000 ms, TI: 880 ms, slice thickness: 1 mm, FoV: 208 × 256 mm, 256 slices, iPAT = 2 |
| VETSA baseline | Siemens | 1.5 | TR = 2730ms, TI = 1000 ms, TE = 3.31ms, slice thickness = 1.33 mm, flip angle = 7°, voxel size 1.3 × 1.0 × 1.3 mm. Acquisition in Boston and San Diego. |
| VETSA follow-up (Boston) | Siemens Tim Trio | 3.0 | TE = 4.33 ms, TR = 2170 ms, TI = 1100 ms, flip angle = 7°, pixel bandwidth = 140, number of slices = 160, slice thickness = 1.2 mm. Acquisition in Boston. |
| VETSA follow-up (San Diego) | GE Discovery 750x | 3.0 | TE = 3.164 ms, TR = 8.084 ms, TI = 600 ms, flip angle = 8°, pixel bandwidth = 244.141, FoV = 24 cm, frequency = 256, phase = 192, number of slices = 172, slice thickness = 1.2 mm. Acquisition in San Diego. |

TR: Repetition time, TE: Echo time, TI: Inversion time, FoV: Field of View, iPat: in-plane acceleration.

results. Also, all reconstructed surfaces were inspected and discarded if they did not pass internal quality control. This led to the exclusion of 46 participants from MoBa-Neurocog and nine from ND, reducing the total LCBC sample to the reported 1633 scans. FreeSurfer 5.3 was used for the LCBC and VETSA analyses, while Lifebrain and UKB MRI data were processed with FreeSurfer 6.0. UKB scans were QC by the UKB imaging team. Further details of the UKB imaging protocol (http://bio-bank.ctsu.ox.ac.uk/crystal/refer.cgi?id=2367) and structural image processing are provided on the UK Biobank website (http://biobank.ctsu.ox.ac.uk/crystal/refer.cgi?id=1977).

## Genetic correlations

Genetic change correlations were obtained by latent change score analyses on the VETSA one and VETSA two subcortical data (see *Brouwer et al., 2017*). All subcortical volumes were adjusted for site and ICV. Left and right volumes at baseline and follow-up for each subject were included in a variant of the 'latent change model' to characterize baseline subcortical volume and change in sub-cortical volume across the two assessments (*McArdle and Plassman, 2009*), with the extension of modeling genetic and environmental effects on the phenotypes (*Panizzon et al., 2015*). The model allows for the estimation of the means and variances of the intercept and slope factors, the relative genetic (i.e., heritability) and environmental contributions to those variances, as well as the pheno-typic, genetic, and environmental correlations between the latent factors. A genetic correlation matrix was generated by estimating genetic correlations of slope factors between all pairwise combi-nations of subcortical structures in bivariate latent change models.

For the UKB SNP analyses, the volume measures of the 16 subcortical structures were corrected for ICV and normalized to have zero mean and one standard deviation, separately, before estimating genetic correlations. We used the bivariate restricted maximum likelihood methods implemented in the program Genome-wide Complex Trait Analysis (GCTA, *Yang et al., 2011*) to compute the genetic correlation for the volume measures for each pair of the 16 brain subcortical structures, including the first 10 principal components, sex and age as covariates. The likelihood ratio test from GCTA testing whether a genetic correlation is zero was used to compute p-values for estimated genetic correlations.

## Experimental design and statistical analysis

GAMM implemented in R (http://www.r-project.org) using the package 'mgcv' (*Wood, 2006*) was used to derive age trajectories for all structures based on the 1633 LCBC MRIs. Annual symmetrized percent change (APC) in volume was correlated across structures in each sample separately (devel-opment and adult/aging from LCBC and Lifebrain). To identify clusters of correlations that could be compared across matrices, the community structure or modules in the matrices were obtained using the Louvain algorithm (*Blondel et al., 2008b*), part of the Brain Connectivity Toolbox (http://www.brain-connectivity-toolbox.net *Rubinov and Sporns, 2010*). The optimal community structure is a subdivision of the network into non-overlapping groups of regions in a way that maximizes within-group connection strength and minimizes between-group strength. The community structure may vary from run to run due to heuristics in the algorithm pertaining to the order in which the nodes are considered, so 10,000 iterations of the algorithm were run, and each region assigned to the module it was most often associated with (by taking the mode of the module assignment across iterations). Negative values were treated asymmetrically (*Rubinov and Sporns, 2011*). To account for global brain changes, between-regional correlations were de-meaned before they were entered into the clustering analyses.

## Acknowledgements

The Lifebrain project is funded by the EU Horizon 2020 Grant: 'Healthy minds 0–100 years: Optimis-ing the use of European brain imaging cohorts ('Lifebrain')'. Grant agreement number: 732592. Call: Societal challenges: Health, demographic change and well-being. In addition, the different sub-stud-ies are supported by different sources:

LCBC: The European Research Council under grant agreements 283634, 725025 (to AMF), and 313440 (to KBW), as well as the Norwegian Research Council (to AMF, KBW). Betula: a scholar grant from the Knut and Alice Wallenberg (KAW) foundation to LN University of Barcelona: Partially sup-ported by a Spanish Ministry of Science, Innovation and Universities (MICIU/FEDER; RTI2018-

095181-B-C21) to DB-F, which was also supported by an ICREA Academia 2019 grant award.; by the Walnuts and Healthy Aging study (http://www.clinicaltrials.gov; Grant NCT01634841) funded by the California Walnut Commission, Sacramento, California. BASE-II has been supported by the German Federal Ministry of Education and Research under grant numbers 16SV5537/16SV5837/16SV5538/16SV5536K/01UW0808/01UW0706/01GL1716A/01GL1716B, and SK has received support from the European Research Council under grant agreement 677804. Cam-CAN: Initial funding from the Biotechnology and Biological Sciences Research Council (BBSRC), followed by support from the Medical Research Council (MRC) Cognition and Brain Sciences Unit (CBU). VETSA is supported by U.S. National Institute on Aging grants R01s AG022381, and AG050595. Part of the research was conducted using the UK Biobank resource under application number 32048.

## Additional information

### Funding

| Funder | Grant reference number | Author |
|---|---|---|
| European Research Council | 283634 | Anders Martin Fjell<br>Kristine Beate Walhovd |
| Horizon 2020 | 732592 | Kristine Beate Walhovd |
| Knut and Alice Wallenberg Foundation | | Lars Nyberg |
| Norwegian Research Council | | Anders Martin Fjell<br>Kristine Beate Walhovd |
| Spanish Ministry of Science and Innovation | MICIU/FEDER/RTI2018-095181-B-C21 | David Bartres-Faz |
| California Walnut Commission | NCT01634841 | David Bartres-Faz |
| Federal Ministry of Education and Research | 16SV5537/16SV5837/16SV5538/16SV5536K/01UW0808/01UW0706/01GL1716A/01GL1716B | Ulman Lindenberger |
| European Research Council | 677804 | Simone Kühn |
| Biotechnology and Biological Sciences Research Council | | Rogier Andrew Kievit |
| Medical Research Council | | Rogier Andrew Kievit |
| U.S. National Institute on Aging | AG022381 | William S Kremen |
| European Research Council | 725025 | Anders Martin Fjell<br>Kristine Beate Walhovd |
| European Research Council | 313440 | Anders Martin Fjell<br>Kristine Beate Walhovd |
| U.S. National Institute on Aging | AG050595 | William S Kremen |
| Institució Catalana de Recerca i Estudis Avançats | ICREA Academia-2019 | David Bartres-Faz |

The funders had no role in study design, data collection and interpretation, or the decision to submit the work for publication.

### Author contributions

Anders Martin Fjell, Conceptualization, Resources, Data curation, Software, Formal analysis, Funding acquisition, Investigation, Visualization, Methodology, Writing - original draft, Project administration, Writing - review and editing; Hakon Grydeland, Formal analysis, Investigation, Methodology, Writing - review and editing; Yunpeng Wang, Data curation, Formal analysis, Investigation, Methodology, Writing - review and editing; Inge K Amlien, David Bartres-Faz, Sandra Düzel, Carol E Franz, Tim C Kietzmann, Stine K Krogsrud, Didac Macía, Athanasia Monika Mowinckel, Cristina Solé-Padullés,

Data curation, Writing - review and editing; Andreas M Brandmaier, Rogier Andrew Kievit, Simone Kühn, Ulman Lindenberger, Rene Westerhausen, Writing - review and editing; Jeremy Elman, Data curation, Formal analysis, Methodology, Writing - review and editing; Asta K Håberg, Data curation, Project administration, Writing - review and editing; William S Kremen, Data curation, Funding acquisition, Project administration, Writing - review and editing; Lars Nyberg, Funding acquisition, Project administration, Writing - review and editing; Matthew S Panizzon, Data curation, Formal analysis, Supervision, Methodology, Writing - review and editing; Øystein Sørensen, Formal analysis, Methodology, Writing - review and editing; Kristine Beate Walhovd, Conceptualization, Supervision, Funding acquisition, Investigation, Project administration, Writing - review and editing

### Author ORCIDs

Anders Martin Fjell (iD) https://orcid.org/0000-0003-2502-8774
David Bartres-Faz (iD) https://orcid.org/0000-0001-6020-4118
Andreas M Brandmaier (iD) https://orcid.org/0000-0001-8765-6982
Rogier Andrew Kievit (iD) http://orcid.org/0000-0003-0700-4568
Simone Kühn (iD) http://orcid.org/0000-0001-6823-7969
Ulman Lindenberger (iD) http://orcid.org/0000-0001-8428-6453
Athanasia Monika Mowinckel (iD) http://orcid.org/0000-0002-5756-0223
Lars Nyberg (iD) http://orcid.org/0000-0002-3367-1746

### Ethics

Human subjects: The studies were approved by the Norwegian Regional Committee for Medical and Health Research Ethics South. Written informed consent was obtained from all participants older than 12 years of age and from a parent/guardian of volunteers under 16 years of age. Oral informed consent was obtained from all participants under 12 years of age. Non-Norwegian samples were approved by the relevant ethical review board for each country. Norway (2010/2359; 2010/3407; 2009/200).

### Decision letter and Author response

Decision letter https://doi.org/10.7554/eLife.66466.sa1
Author response https://doi.org/10.7554/eLife.66466.sa2

## Additional files

### Supplementary files

- Source code 1. Statistical source code.
- Supplementary file 1. Supplementary results.
- Transparent reporting form

### Data availability

The study comprises many different data sources. The PI does not have the legal right to share these data directly. UK Biobank data can be obtained from http://www.ukbiobank.ac.uk. The data repository for the Cambridge Centre for Ageing and Neuroscience (Cam-CAN) dataset can be found at http://www.cam-can.org/index.php?content=dataset. Access to BASE-II data can be obtained at http://www.base2.mpg.de/7549/data-documentation. Access to VETSA data can be obtained at https://medschool.ucsd.edu/som/psychiatry/research/VETSA/Researchers/Pages/default.aspx. Betula is described at http://www.umu.se/en/research/projects/betula—aging-memory-and-dementia/. For data from Barcelona brain studies, see http://www.neurociencies.ub.edu/david-bartres-faz/. For LCBC Lifespan sample, contact information can be found at https://www.oslobrains.no/presentation/anders-m-fjell/. Part of the developmental sample can be accessed through https://www.fhi.no/en/studies/moba/for-forskere-artikler/research-and-data-access/ (As of 2021, we are in the process of transferring MRI data to this repository). Please note that for all samples, data transfer agreements must be signed and proper ethical and data protection approvals must be in place, according

to national legislation. Code used for data analysis accompany the submission as separate files. The correlation matrices constituting the basis for the Mantel tests are also uploaded.

The following datasets were generated:

| Author(s) | Year | Dataset title | Dataset URL | Database and Identifier |
|---|---|---|---|---|
| Kievit RA | 2021 | Cam-CAN | https://www.cam-can.org/index.php?content=dataset | Cam-CAN Data Repository, www.cam-can.org/index.php?content=dataset |
| Düzel S, Lindenberger U, Kühn S | 2021 | Berlin Aging Study II | https://www.base2.mpg.de/7549/data-documentation | BASE-II Data, data-documentation |
| Franz CE, Kremen WS | 2021 | VETSA | https://medschool.ucsd.edu/som/psychiatry/research/VETSA/Researchers/Pages/default.aspx | UC San Diego School of Medicine, VETSA |
| Nyberg L | 2021 | BETULA | https://www.umu.se/en/research/projects/betula—aging-memory-and-dementia/ | Umeå University, aging-memory-and-dementia/ |
| Bartres-Faz D, Solé-Padullés C | 2021 | Barcelona Brain studies | http://www.neurociencies.ub.edu/david-bartres-faz/ | The Institute of Neurosciences, www.neurociencies.ub.edu/david-bartres-faz/ |
| Fjell AM, Walhovd KB | 2021 | LCBC brain studies | https://www.oslobrains.no/presentation/anders-m-fjell/ | LCBC, www.oslobrains.no/presentation/anders-m-fjell/ |
| Fjell AM, Walhovd KB | 2021 | MoBa neurocognitive study | https://www.fhi.no/en/studies/moba/for-forskere-artikler/research-and-data-access/ | NIPH, data-access/ |

The following previously published dataset was used:

| Author(s) | Year | Dataset title | Dataset URL | Database and Identifier |
|---|---|---|---|---|
| Smith SM | 2020 | UK Biobank | https://www.ukbiobank.ac.uk/enable-your-research/about-our-data/imaging-data | imaging-data, UKBiobank |

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
