## [Decision Letter]

**Acceptance summary:**

This paper makes an important contribution to understanding human brain development and will be of note to readers with a particular interest in subcortical anatomy. The work sheds new light on principles of subcortical maturation and its genetic mechanisms.

**Decision letter after peer review:**

Thank you for submitting your article "The genetic organization of longitudinal subcortical volumetric change is stable throughout the lifespan" for consideration by *eLife*. Your article has been reviewed by 3 peer reviewers, one of whom is a member of our Board of Reviewing Editors, and the evaluation has been overseen by Tamar Makin as the Senior Editor. The reviewers have opted to remain anonymous.

The Reviewers were all enthusiastic about the potential of the paper, but had a number of methodological concerns that will be important to address. We would be happy to consider a revision provided that you are able to comprehensively address all the points raised.

Essential revisions:

1. The analysis includes data from a wide age range spanning 4 to 88 years. T1 signal characteristics, particularly those affecting tissue contrast, can change significantly over this time. This will have an impact on the accuracy of any automated segmentation algorithm. How can we be sure that age-related differences are not simply due to variations in tissue contrast? This problem is compounded by the reliance on the Freesurfer aseg algorithm, which parcellates the brain using an adult training set. Thus, not only will there be a problem of age-related differences in tissue contrast, but also in the accuracy with which individual T1s can be spatially aligned to the template, which is likely to decline as a function of age difference from the young adults used to generate the templates/training set. The authors should demonstrate that such effects cannot explain their findings.

2. The authors conclude that "at a general level, change in the structures tended to cluster according to trends from embryonic development and placement along the cranial vertical axis, but with notable exceptions.". This is based on the unsupervised clustering solution from correlating inter individual variation in change for each structure (which they argue is reproduced in development aging and genetic analyses – see point 2 below). Looking at their clustering solution though, a more superficial explanation could easily explain the 3 main clusters they observe: Cluster 1 = fluid-filled ventricles, Cluster 2 = White matter (or white matter rich for thalamus, hippocampus and cerebellar cortex) ROIs, Cluster 3 = Most other ROIs, which are general non-white matter rich cortical or subcortical nuclei. I understand that linkage to embryological patterning is more profound – but there must be some effort to address the more concrete possibility that the clustering is basically saying CSF goes with CSF, white with white, and gray with gray (put crudely). Although thalamus, hippocampus and cerebellar cortex are classed as gray by tissue classification algorithms – it is well recognized that these structures also contain substantial white matter components. We suggest comparing mean MT or FA for Clusters 1 and 2 to get at this empirically. Evidence for the "tissue composition" hypothesis for clustering is not incompatible with the "embryological origins" hypothesis – but focusing on the latter to the extent that the authors do would be more justified if there were not evidence of the former.

3. The authors conclude that their findings show "substantial coherence in the pattern of change between development and the rest of the life". This major finding is based mainly on the cross-cell correlations and Mantel tests for compare different correlation matrices: (i) correlated intra-individual changes in development, (ii) correlated changes in adult/aging, (iii) genetic correlation in change, and (iv) SNP-based estimated of co-heritability. The dominant signal by far within any one of these matrices, and the main signal that appears to be recovered across matrices is the contrast between ventricular measures and non-ventricular measures. The reproducibility of this contrast is less meaningful than the reproducibility e.g. based on correlation across all non-ventricle edges between matrices. First and foremost, a fundamental issue with giving equal weight to ventricular vs. non-ventricular measures in their contribution to statements of similarity between matrices is that the ventricles are a single continuous fluid filled cavity, so that saying – eg lateral and lateral inf ventricles are correlated is considerably different from saying amygdala and thalamus are. Compounding this concern is that the weight given the ventricle-related edges in driving cross-matrix correspondences is somewhat arbitrarily defined by how many parts you want to chop the ventricles into. This level of arbitrariness does not apply for example for the distinction between the other main ROIs included. To address these issues, the cross-edge correlation between matrices should be presented and probed through scatterplots. Scatterplots for these edge-wise correlations between matrices should lay out the issue (using edge here to mean cell in the matrix). The ~0.8 value is likely to be inflated by dense point of edges to the top right and bottom left of the plot relating to inter ventricular edges and ventricle-nonCSF edges respectively. These scatterplots between matrices should be visualized with all edges, then with just the edges that do not include ventricles. The correlation should be recalculated for the latter, and a fit line shown to help readers identify those edges that show weakest correspondence between matrices. A v efficient way of looking at the sensitivity tests above is constructing a symmetric 5*5 matrix – each column/row is a matrix from Figures 2 and 4. The cells show inter-matrix correlations. Show this one with ventricles in, and once with them out. The core issue here is showing that the signals being discussed and interpreted aren't overly dependent in the simple contrast between tissue and non-tissue compartments.

4. The Mantel test is likely biased in the case of spatially autocorrelated data (Guillot and Rousset, 2013, Methods in Ecol and Evol). The authors should consider generating spatially constrained null models (e.g., Alexander-Bloch et al. NeuroImage, 2018; Burt et al. NeuroImage, 2020) for inference on the Mantel test.

5. It may be, in part, due to how you phrase some points but you seem to suggest that volumetric correlation is analogous to volumetric change correlation, which is not supported by the analyses you perform. The fact that the matrices of correlations between volume and volume change resemble each other does not prove they are driven by similar (genetic) factors. You would need to evaluate correlation between volume and volumetric change, in particular at a genetic level to support this claim. Here is a selection of the sentences I found misleading or problematic:

This was confirmed by single nucleotide polymorphisms-based analyses of 38127 cross-sectional MRIs.

Importantly, both the structure and the coordinated change of each cluster tended to be governed by common sets of genes

This means that regions that develop and change together through life tend to be governed by the same sets of genes

Rather, genetic influence on change rates and baseline volume overlapped for most structures. This finding allowed us to use cross-sectional data from UKB to further explore the genetic contribution to subcortical organization.

The similarity of the developmental change matrix and the SNP genetic correlation matrix obtained from middle-aged adults thus yielded further support for the hypotheses that genetically governed neurodevelopmental processes can be traced in subcortical structures through life.

6. You also seem to assume that the clusters defined from pairwise correlations would consist in homogeneous regions (e.g. with common genetics or determinants). I would argue that this is not necessarily the case, and it could be investigated at a genetic level using multivariate twin models (e.g. common pathway model).

7. I would find useful that you highlight the significant phenotypic and genetic correlations (e.g. in Figure 2 and 4). For example, I would find a cluster consisting of significant correlations more convincing. On the other hand, a significant positive correlation between clusters would suggest that the clusters are not independent, which is also interesting.

8. In my opinion, the description of age trajectories, albeit extremely interesting, are not central to the scientific question you are trying to tackle. I think it would improve readability to present them after the main results in the same section as the cluster trajectories. In particular, reading the result section led me to think that GAMM were central to your analysis, while they are only used for a visualisation of the age trajectories, which only serves a descriptive purpose.

9. More generally, the result section would benefit from being more linear (starting from the main results) and/or would benefit from a better correspondence between the figures and the text. At the moment, it feels like the figures and text have different progressions all together, which is confusing. For example, the results presented in Figure 2 are spread in different sections of the text that are not even following each other. Figure 4 would make sense near Figure 2, or even as an additional panel of Figure 2. An example of what I think would be a more linear progression of the results.

o E.g. Phenotypic correlations/clusters.

o Replication in adults.

o Sensitivity analysis.

o Genetic correlations clusters.

o Relationship with embryonic development.

o Visualisation of age trajectories of regions and clusters.

o Additional analyses (e.g. cognition).

10. Modularity analysis – There does not appear to be any assessment of the statistical significance of the cluster solution. This could be done by shuffling the correlation matrix using appropriate algorithms, depending on the degree to which the authors want to consider spatial effects in the data (e.g., Roberts et al. 2015, NeuroImage; Rubinov and Sporns, NeuroImage, 2011).

11. Modularity analysis – it appears the authors used a standard implementation of the Louvain algorithm. The default definition of Q defines a null expectation for within-cluster connectivity that is not appropriate for correlation matrices. With such data, the mean correlation can be appropriate.

12. Modularity analysis – the evaluation of different levels of γ is commendable. It is unclear why this is left to the end of the Results. This should either be placed at the beginning, as a way of justifying the 5-cluster solution, or the beginning should indicate what value of γ was used initially and contain a reference to subsequent analyses that investigate the issue of γ in more detail.

13. The similarity between the clusters and developmental origins is qualitative. Given that this is a core aim of the analysis, perhaps the authors could perform some inference by shuffling the labels and using a measure of partition similarity such as their normalized mutual information?

14. Please explain why the cortex is treated as a single structure? Cortical development is regionally heterogeneous.

15. Sex is known to modify trajectory shape for cortical and subcortical structures. It would be good to show that the observed clustering holds for male and female subsamples. This would provide an important sensitivity analysis and also potential evidence for a sort of split half reliability.

16. It would also be important to show how ROIs cluster when you use the first derivatives from the gam fits in Figure 1. That would provide a complimentary approach to the inter-individual change method on which current work is built, and also help unpack some of the potential concerns re "fluid vs. tissue" and "white right vs. other" as two big potential drivers for findings. I would add plots for the ventricular components in Figure 1 too, as well as dendrograms for the primary heat map clustering used to order other matrices. My guess would be split one is CSF vs. tissue, and split 2 is white/white-rich tissue vs. other tissue.

17. In estimating APCs – how did the authors deal with were people with more than 2 scans, and the potential for including mid-scan age as a factor before correlating given the non-linear volume changes in development?

18. I may have missed it but I did not expect the analysis of cognitive function. What purpose it is serving and does it really integrate in the theory of neurodevelopment? In addition, only one association seems significant after multiple testing and the difference seems to be localised in the 25-60 years old group for which you have the fewest observations (from Figure 1). Could this be due to a handful of outliers? Could you also add precision about which test was used? Maybe reporting several curves for difference quantiles of the distribution (instead of above/below average) would help visualise the effect?

19. Figures 4 and 5 need a color scale.

20. Figure 2 is the main figure of result and I strongly suggest you expand the caption to improve self-readability. For example, the fact that the first row are results obtained on the LCBC sample. Clarify that the clustering was not recalculated for each matrix but is that of the first panel. In addition, the titles of the correlation plots could be clearer (e.g. clearly stating phenotypic and genetic correlations). Also the 4th panel is not a volumetric change-change relationship, which is the title of the figure.

21. Abstract and introduction should announce that the focus is about volume of subcortical structures.

22. The discussion reads well but contains many statements that do not seem supported by the results or that sounded too definitive considering the analysis only focused on 16 volumetric measurements. It also lacks a limitation section.

23. Some aspects of the text could be clearer. For example:

Line 44 – it is unclear what the hypothesis is precisely.

Line 59 – what does a topographic organization correspond to in this context? regionally specific?

Line 61 – it is unclear what it means to follow the "genetic organization of the cortex".

Line 79 – it is very hard to understand what this hypothesis refers to and what it predicts.

Line 88 – it is unclear what a genetic anatomical architecture is.

Line 105 – please clarify: was the correlation across subjects?

Line 121 – it is unclear what "common factors" refers to here.

Line 217 – only a correlation rather than prediction has been shown.

Line 256 – please clarify: was consensus clustering run over 1000 runs at each γ?

“Although the lifespan trajectories of subcortical structures are much more divergent than those for cortical regions.”

Unclear to me how this is supported by results from the article, could you expand on how you conclude this?

“Mapping the developmental clusters to the adult part of the sample yielded highly different change trajectories.”

Here it may help to cite figures and tables in Discussion – also is this the case for all clusters?

“Clusters 1, 3 and to a lesser extent 4 were related to general cognitive function in a mostly age-invariant manner, which implies that the relationship between cognitive function and subcortical volumes is established early in life.”

This sounds like a huge overstatement. You studied only volume change of 16 regions – and the associations with cognition scores are not the most convincing.

“This means that regions that develop and change together through life tend to be governed by the same sets of genes.”

Beyond my previous criticism, this reads as a rather large generality.

“…and are governed by distinct sets of genes.”

What is this conclusion based on?

“It has also been argued that genes expressed in the subcortex generally are more region-specific and tend to evolve more rapidly than genes expressed in cortical regions”.

I am not sure what you mean by "evolve more rapidly" is it that it is under greater selection pressure?

The SNP genetic correlation matrix was highly similar to the developmental change matrix, as demonstrated by the Mantel test (r = 0.57, p <.0005, see Figure 2).”

I am not a big fan of qualitative judgments (e.g. highly). Especially that I assume r can vary between 0 and 1 so "highly" can be seen as an overstatement.

As much as the Mantel test statistics are different from 0, are they also different from 1?

Generalized

“Additive Mixed Models (GAMM)”

Could you add some details about the maximal order of the splines you considered? How was the best model – best order selected?

“This revealed a close to perfect match between the adult genetic clusters and their embryonic origins.”

24. You say grouping is extremely similar, but compared to the clusters presented in Figure 2, I feel that this is an overstatement. E.g. caudate, accumbens, putament, pallidum – two of those were in separate clusters previously. Also positive rG between cerebellum cortex and WM while this was negative before.

25. “In all cases yielded the slope function the lowest IC values”. This is not true – see caudate and cortex.

What to make from small AIC/BIC differences i.e. a marginally better fit? I wonder whether Table 3 is really adding anything, especially that the age trajectories are mostly descriptive?

26. “We fitted the developmental trajectory of each cluster…”.

I assume you took the total volume of each cluster? What is this analysis really adding? First it assumes the cluster is somewhat homogenous (see my second comment) and another problem is that the different volumes can have extremely different scales which makes interpreting the sum difficult.

27. “Using multivariate latent change score models, we calculated…”.

Why did you not also use the annual symmetrized percent change you used previously? Especially that you want to compare rG to the phenotypic correlations you previously studied.

28. rGs from twin models are corrected for ICV, which does not seem to be the case for the GREML approach. Is this simply missing in the text?

“…pair of the 16 brain sub-cortical structures, including the first ten principal components, sex and age as covariates.”

29. “log likelihood test”. I am more used to it being referred to as the likelihood ratio test.

30. “restricted maximum likelihood methods” – (very) minor detail but REML is the optimisation approach used to estimate the parameters of what is a bivariate mixed model. A compromise may be to talk about GREML, which is not more correct but quite commonly used to refer to the LMM implemented in GCTA. For an analogy, it is as if you referred to the GAMM as a Restricted Marginal Likelihood method.

31. “…due to heuristics in the algorithm…”. What are the heuristics here, is it random starting values?

32. “Also, all reconstructed surfaces were inspected, and discarded if they did not pass internal quality control.” Surely, this does not apply to the UKB analyses. Did you perform any QC on them beyond the UKB provided screening?

33. “In addition, we removed participants suggested to be removed for genetic analysis by the UK Biobank team.” Could you add a precision about why this exclusion is recommended?

---

## [Author Response]

Essential revisions:1. The analysis includes data from a wide age range spanning 4 to 88 years. T1 signal characteristics, particularly those affecting tissue contrast, can change significantly over this time. This will have an impact on the accuracy of any automated segmentation algorithm. How can we be sure that age-related differences are not simply due to variations in tissue contrast? This problem is compounded by the reliance on the Freesurfer aseg algorithm, which parcellates the brain using an adult training set. Thus, not only will there be a problem of age-related differences in tissue contrast, but also in the accuracy with which individual T1s can be spatially aligned to the template, which is likely to decline as a function of age difference from the young adults used to generate the templates/training set. The authors should demonstrate that such effects cannot explain their findings.

We fully agree that there are substantial age-differences in T1 signal intensities and tissue contrast. In previous work, we have shown that GM/WM contrast in T1w images is lower in older than younger participants (Increased sensitivity to effects of normal aging and Alzheimer's disease on cortical thickness by adjustment for local variability in gray/white contrast: A multi-sample MRI study – ScienceDirect) and that T1w signal intensity is related to chronological age both within the cortex and in subcortical structures (Differentiating maturational and aging-related changes of the cerebral cortex by use of thickness and signal intensity – PubMed (nih.gov)). GM/WM contrast changes can reliably be measured longitudinally, and the rate of contrast decay seems related to the initial regional myelin content (Accelerated longitudinal gray/white matter contrast decline in aging in lightly myelinated cortical regions – Vidal‐Piñeiro – 2016 – Human Brain Mapping – Wiley Online Library). Almost certainly, such contrast differences will affect brain segmentations, as we showed in the first paper referenced above, and as we discuss in more detail in a developmental context in a separate paper (Through Thick and Thin: a Need to Reconcile Contradictory Results on Trajectories in Human Cortical Development | Cerebral Cortex | Oxford Academic (oup.com)). In short, we found that rather than contrast differences inflating structural age-relationships, correcting for them increased age differences and sensitivity to detect AD. Thus, it is highly unlikely that the reported age-related differences are inflated as a function of contrast and intensity differences. However, importantly, as with all in vivo imaging methods, our study provides merely representations of the underlying neurobiology and inherently require some level of interpretation. It is critical to be aware of the fact that the MRI-derived measures are merely our best current approximations, where segmentations are based on signal intensities and contrast properties that are prone to the influence of multiple factors, including, but not limited to, age. We have previously suggested that the term “apparent cortical thickness” should replace “cortical thickness”, and the same could apply to any MRI-segmented brain volume. Underlying mechanisms of volume differences and change are complex and may involve events such as growth, proliferation of dendrites, dendritic spines, axonal sprouting, vascular elaboration, synaptic pruning as well as myelination. Many of these will likely affect both contrasts, signal intensities and volumetric estimations, but the relative effect of each is challenging to tease apart.

On this background, it is very interesting that we find high stability in the organization of change-change patterns in development and aging. It may be that contrast changes are relatively minor since analyses are based on within-subject longitudinal changes covering only a few years. For the same reasons, effects of inter-subject registrations to the probabilistic atlas have less impact since the used metric is within-subject change.

We agree that these issues are interesting, and that they deserve a more in-depth discussion in the manuscript. We have added the following paragraph to the Discussion section:

“Limitations: Caveats in interpreting brain changes from MRIs and further research

Similar to all studies based on in vivo imaging methods, this study provides approximations of the underlying neurobiology. […] A promising avenue for further research is to use multi-modal neuroimaging with different MRI sequences and analysis methods to yield more insight in the foundations for the volumetric changes studied.”

2. The authors conclude that "at a general level, change in the structures tended to cluster according to trends from embryonic development and placement along the cranial vertical axis, but with notable exceptions.". This is based on the unsupervised clustering solution from correlating inter individual variation in change for each structure (which they argue is reproduced in development aging and genetic analyses – see point 2 below). Looking at their clustering solution though, a more superficial explanation could easily explain the 3 main clusters they observe: Cluster 1 = fluid-filled ventricles, Cluster 2 = White matter (or white matter rich for thalamus, hippocampus and cerebellar cortex) ROIs, Cluster 3 = Most other ROIs, which are general non-white matter rich cortical or subcortical nuclei. I understand that linkage to embryological patterning is more profound – but there must be some effort to address the more concrete possibility that the clustering is basically saying CSF goes with CSF, white with white, and gray with gray (put crudely). Although thalamus, hippocampus and cerebellar cortex are classed as gray by tissue classification algorithms – it is well recognized that these structures also contain substantial white matter components. We suggest comparing mean MT or FA for Clusters 1 and 2 to get at this empirically. Evidence for the "tissue composition" hypothesis for clustering is not incompatible with the "embryological origins" hypothesis – but focusing on the latter to the extent that the authors do would be more justified if there were not evidence of the former.

We agree with the reviewers that tissue composition contributes to the observed clustering, and also that this is not incompatible with the embryological origin hypothesis we have used to guide the interpretation of the results. Myelin content is almost certainly a factor that impacts the clustering. We acknowledged this in the original submission also (Discussion):

“This cluster is also characterized by relatively high myelin content among several of its constituents, which may have contributed to the inclusion of hippocampus and cortical WM.”

We have now expanded on this important point (Discussion):

“The present results reflect effects of various neurobiological events on signal intensities and contrast. […] Thus, the clustering results will likely partly reflect different myelin content in the structures analyzed, as changes is myelin may be correlated across regions in the brain.”

We appreciate the reviewers’ suggestion of running additional analyses by use of MT or FA. However, we believe the results of these will be challenging to interpret with regard to myelin. Although FA probably is related to myelin content, it is a nonspecific measure since there is not a direct correspondence between an FA value and a WM cellular component. For instance, regardless of myelin, high axon packing density and low axon diameter translate into high FA values due to high membrane density perpendicular to the axon. Further, high FA is seen in major tracts in newborns almost without cerebral myelin, and classic studies in rodents found reductions of only about 20% in FA in absence of myelin (shiver mice). Thus, if FA values were higher in cluster 2 than cluster 3, it would be difficult to ascribe this to myelin content. And vice versa, if there were no differences in FA values, this would still not rule out the tissue composition account. Similar considerations regard other MRI-based in vivo measures of white matter and myelin, such as MT, which measures myelination indirectly and can be influenced by for instance water content and neuroinflammation. This was also the main conclusion in a recent *eLife* paper (An interactive meta-analysis of MRI biomarkers of myelin | *eLife* (elifesciences.org)):

“Similarly to other qMRI biomarkers, MRI-based myelin measurements are indirect, and might be affected by other microstructural features, making the relationship between these indices and myelination noisy.”

Regardless of these caveats, we agree with the reviewer that combining different imaging modalities has potential to yield interesting information. We have therefore included a sentence stating this in the revised manuscript (Discussion):

“A promising avenue for further research is to use multi-modal neuroimaging with different MRI sequences and analysis methods to yield more insight in the foundations for the volumetric changes studied.”

Regarding CSF, please see comments to the point below.

3. The authors conclude that their findings show "substantial coherence in the pattern of change between development and the rest of the life". This major finding is based mainly on the cross-cell correlations and Mantel tests for compare different correlation matrices: (i) correlated intra-individual changes in development, (ii) correlated changes in adult/aging, (iii) genetic correlation in change, and (iv) SNP-based estimated of co-heritability. The dominant signal by far within any one of these matrices, and the main signal that appears to be recovered across matrices is the contrast between ventricular measures and non-ventricular measures. The reproducibility of this contrast is less meaningful than the reproducibility e.g. based on correlation across all non-ventricle edges between matrices. First and foremost, a fundamental issue with giving equal weight to ventricular vs. non-ventricular measures in their contribution to statements of similarity between matrices is that the ventricles are a single continuous fluid filled cavity, so that saying – eg lateral and lateral inf ventricles are correlated is considerably different from saying amygdala and thalamus are. Compounding this concern is that the weight given the ventricle-related edges in driving cross-matrix correspondences is somewhat arbitrarily defined by how many parts you want to chop the ventricles into. This level of arbitrariness does not apply for example for the distinction between the other main ROIs included. To address these issues, the cross-edge correlation between matrices should be presented and probed through scatterplots. Scatterplots for these edge-wise correlations between matrices should lay out the issue (using edge here to mean cell in the matrix). The ~0.8 value is likely to be inflated by dense point of edges to the top right and bottom left of the plot relating to inter ventricular edges and ventricle-nonCSF edges respectively. These scatterplots between matrices should be visualized with all edges, then with just the edges that do not include ventricles. The correlation should be recalculated for the latter, and a fit line shown to help readers identify those edges that show weakest correspondence between matrices. A v efficient way of looking at the sensitivity tests above is constructing a symmetric 5*5 matrix – each column/row is a matrix from Figures 2 and 4. The cells show inter-matrix correlations. Show this one with ventricles in, and once with them out. The core issue here is showing that the signals being discussed and interpreted aren't overly dependent in the simple contrast between tissue and non-tissue compartments.

We agree that different CSF compartments may be expected to correlate, probably due to general developmental growth or age-related atrophy, and this is also demonstrated by the relatively high change-change correlations for the different ventricular/ CSF variables. The analyses were also run without inclusion of the ventricles, and although the r-values were reduced, the matrices were still significantly more similar than what would have been expected by chance. To test how the clustering was affected by not including CSF, the following analyses were presented in the manuscript (Results section):

“Third, as changes in different parts of the ventricular system were expected to be highly correlated, we re-ran the Mantel tests excluding the ventricles. […] This was in line with the higher within-cluster than between-cluster correlations reported above for the non-CSF clusters.”

Thus, the reviewer is right in assuming that CSF compartments affected the similarities between the clusters, but the compared change-change matrixes are still significantly more similar than expected by chance when the CSF compartments are not included in the matrix comparisons. We also ran the clustering in the UKB SNP co-heritability matrix with and without CSF compartments included:

“A two cluster solution yielded a trivial divide between a ventricular cluster and one cluster containing the remaining structures. Thus, we ran a separate analysis on the non-ventricular structures. This revealed a match between the adult genetic clusters and their embryonic origins (Figure 3).”

Figure 3 also shows the cluster solution for this analysis not including CSF compartments. Finally, in the revised manuscript, we also describe the three-cluster solution. The following is added to the manuscript (Results, section on Cluster stability analyses):

“In the 3-cluster solution, one cluster consisted of the ventricles, one consisted of pallidum, amygdala and accumbens, while the remaining structures were included in the last cluster.”

Thus, we strongly believe the signals are not merely reflecting the simple contrast between tissue and non-tissue compartments.

4. The Mantel test is likely biased in the case of spatially autocorrelated data (Guillot and Rousset, 2013, Methods in Ecol and Evol). The authors should consider generating spatially constrained null models (e.g., Alexander-Bloch et al. NeuroImage, 2018; Burt et al. NeuroImage, 2020) for inference on the Mantel test.

We see the reviewer’s point. A challenge in our data, however, is that GM and WM cerebrum and cerebellum are included, in which distance measuring becomes problematic as such spatially extended structures do not lend themselves well to either three-dimensional Euclidean distance (subcortical) or surface-based geodesic distance (Burt et al. NeuroImage, 2020). We therefore consider it impractical to generate spatially constrained null models without deviating excessively from the original structures. To take this limitation of the Mantel test into account, in the revised manuscript we have added alternative analyses following Betzel and Bassett (2017) to compare the matrices, not vulnerable to spatial constraints (please see response to point 10 below).

5. It may be, in part, due to how you phrase some points but you seem to suggest that volumetric correlation is analogous to volumetric change correlation, which is not supported by the analyses you perform. The fact that the matrices of correlations between volume and volume change resemble each other does not prove they are driven by similar (genetic) factors. You would need to evaluate correlation between volume and volumetric change, in particular at a genetic level to support this claim. Here is a selection of the sentences I found misleading or problematic:This was confirmed by single nucleotide polymorphisms-based analyses of 38127 cross-sectional MRIs.Importantly, both the structure and the coordinated change of each cluster tended to be governed by common sets of genesThis means that regions that develop and change together through life tend to be governed by the same sets of genesRather, genetic influence on change rates and baseline volume overlapped for most structures. This finding allowed us to use cross-sectional data from UKB to further explore the genetic contribution to subcortical organization.The similarity of the developmental change matrix and the SNP genetic correlation matrix obtained from middle-aged adults thus yielded further support for the hypotheses that genetically governed neurodevelopmental processes can be traced in subcortical structures through life.

We did not intend to imply that volumetric correlations are analogous to volumetric change correlations. Except for the UKB SNP-analyses, which are based on volume-correlations, all analyses in the paper are longitudinal (change). Unfortunately, the longitudinal UKB SNP analyses did not yield stable results due to lack of power (Results):

“In order to further explore the genetic contributions to coordinated subcortical change, we first attempted to calculate the pairwise single nucleotide polymorphism (SNP)-based genetic correlation between change in each pair of structures by running a mega-analysis on 1337 participants with longitudinal MRIs from UK Biobank and 508 from LCBC. […] Thus, we instead based the SNP genetic analyses on the cross-sectional UKB data where power is much greater (n = 38127, age 40-69 years), using age, sex and the first 10 components of the genetic ancestry factor as covariates.”

Thus, UKB results are cross-sectional only, and statements about genetic contributions to the change correlations are based on the twin analyses, for which power is greater to detect genetic contributions to change compared with SNP-bases analyses.

To avoid misunderstandings, we have re-formulated the statements highlighted by the reviewers as confusing:

“This was confirmed by single nucleotide polymorphisms-based analyses of 38127 cross-sectional MRIs.” “Single nucleotide polymorphisms-based analyses of 38127 cross-sectional MRIs showed a similar pattern of genetic volume-volume correlations.”

“Importantly, both the structure and the coordinated change of each cluster tended to be governed by common sets of genes” “Importantly, both the volumetric correlations within each cluster and the coordinated change of each cluster tended to be governed by common sets of genes.”

“This means that regions that develop and change together through life tend to be governed by the same sets of genes” “We found that longitudinal volumetric change in regions that cluster together are influenced by the same genes. […] Further, the SNP genetic correlation analysis showed that also the cross-sectional volumetric correlations followed a similar organization.”

“Rather, genetic influence on change rates and baseline volume overlapped for most structures. This finding allowed us to use cross-sectional data from UKB to further explore the genetic contribution to subcortical organization.” “Thus, we used cross-sectional data from UKB to further explore the genetic contribution to subcortical volumetric organization. Although the cross-sectional nature of these data prevents conclusions about change-change relationships per se, they increased sample size for these analyses from 6000 to more than 38000 MRIs.”

“The similarity of the developmental change matrix and the SNP genetic correlation matrix obtained from middle-aged adults thus yielded further support for the hypotheses that genetically governed neurodevelopmental processes can be traced in subcortical structures through life.” “The similarity of the developmental change matrix and the SNP genetic volume-volume correlation matrix obtained from middle-aged adults thus yielded further support for the hypotheses that genetically governed neurodevelopmental processes can be traced in subcortical structures through life.”

6. You also seem to assume that the clusters defined from pairwise correlations would consist in homogeneous regions (e.g. with common genetics or determinants). I would argue that this is not necessarily the case, and it could be investigated at a genetic level using multivariate twin models (e.g. common pathway model).

We agree with the reviewer that the clusters will not consist of homogenous regions. What we attempt to show is that there are meaningful clusters that can be identified, within which change is more highly correlated and genetic influence is shared to a larger degree than what it the case for regions outside the clusters. With this, however, we do not mean to imply that the regions within each cluster are homogenous. To make this clear, we have added a paragraph to the Limitation section of the Discussion:

“A second caveat is that although the clustering of regions is based on pairwise change-change correlations, this does not imply that each cluster consists of homogenous regions. Regions within a cluster show more correlated volumetric change with other regions within the cluster than with regions outside the cluster, and the genetic analyses show higher genetic correlations for change and absolute volume with other within-cluster regions than regions outside the cluster.”

7. I would find useful that you highlight the significant phenotypic and genetic correlations (e.g. in Figure 2 and 4). For example, I would find a cluster consisting of significant correlations more convincing. On the other hand, a significant positive correlation between clusters would suggest that the clusters are not independent, which is also interesting.

We understand the reviewers’ point. However, the focus of the paper is the *patterns* of change, not the pairwise relationships per se. Further, with our sample size, we believe the significance values are less relevant. For instance, in Lifebrain and for UKB genetic correlations, the critical r-value is below 0.075 for p <.05 (two-tailed), and for the cluster forming LCBC development sample, critical r = .10. The majority of the pairwise within-cluster correlations are significant at p <.05 (48 out of 60 for UKB, see supplemental information, 41 out of 60 for LB), but explained variance may be as low as < 1%, and we therefore believe the significance testing of the correlations presented in Table 5 is more appropriate than reporting p-values for each pairwise correlation. The p-values in Table 5 are not dependent on sample size, which is a further complicating issue when reporting p-values: two identical matrices may show substantial differences regarding which correlations are significant and which are not. As the analyses are based on the pattern of correlations, we are afraid that including pairwise significance levels in the figures will be confusing. Thus, we would prefer not to highlight significance levels of the individual cells in figures 1 and 2.

8. In my opinion, the description of age trajectories, albeit extremely interesting, are not central to the scientific question you are trying to tackle. I think it would improve readability to present them after the main results in the same section as the cluster trajectories. In particular, reading the result section led me to think that GAMM were central to your analysis, while they are only used for a visualisation of the age trajectories, which only serves a descriptive purpose.

Thank you for this suggestion. We have now moved the description of the age trajectories as suggested by the reviewer, and present them after the cluster trajectories. We have also completely re-ordered the Results section according to the reviewer request #9 below.

9. More generally, the result section would benefit from being more linear (starting from the main results) and/or would benefit from a better correspondence between the figures and the text. At the moment, it feels like the figures and text have different progressions all together, which is confusing. For example, the results presented in Figure 2 are spread in different sections of the text that are not even following each other. Figure 4 would make sense near Figure 2, or even as an additional panel of Figure 2. An example of what I think would be a more linear progression of the results.o E.g. Phenotypic correlations/clusters.o Replication in adults.o Sensitivity analysis.o Genetic correlations clusters.o Relationship with embryonic development.o Visualisation of age trajectories of regions and clusters.o Additional analyses (e.g. cognition).

We agree that this is a good idea. We have now re-ordered the whole Results section. We have also replaced Figure 1 and Figure 2 with new figures better suited to illustrate the new organization of the results. Figure 1 now contains all the three phenotypic change-change correlation matrices. Figure 2 contains the genetic matrices. We believe this re-organization has improved the structure of the manuscript considerably.

10. Modularity analysis – There does not appear to be any assessment of the statistical significance of the cluster solution. This could be done by shuffling the correlation matrix using appropriate algorithms, depending on the degree to which the authors want to consider spatial effects in the data (e.g., Roberts et al. 2015, NeuroImage; Rubinov and Sporns, NeuroImage, 2011).

As mentioned above, it is problematic to account for spatial effects in the current data given the inclusion of measures from the entire brain (GM and WM). We therefore tested the statistical significance of the cluster solution by shuffling the correlation matrix without considering the spatial effects. The following additions were made:

Results, section Clusters of change in development:

“Five clusters of coordinated developmental change were identified (Figure 1) (see Cluster stability analyses and Validation analyses below for a more detailed discussion and justification of the cluster solution). […] The community-structure solution was significantly more clustered than in the random networks (p<0.001, developmental change Q=0.44, the 2.5 and 97.5 percentile of the random Q distribution=0.36-0.40).”

Further, as an additional assessment of the cluster solutions, we tested the similarities of the community structure (cluster solution) of the developmental change and the adult/aging change. Following Betzel and Bassett 2017, we calculated the normalized mutual information (Lancichinetti et al., 2009), variation of information (Meilă, 2003), and the z-score of the Rand coefficient (Traud et al., 2011). The following was added to the manuscript (Results, section on Cluster stability analyses):

“Finally, we tested the similarities of the community structure (cluster solution) of the developmental and the adult/aging change-change matrices. […] Due to the nature of the research questions and data, including both GM and WM compartments as single structures, the null models generated were not spatially constrained (Alexander-Bloch et al., 2018; Burt, Helmer, Shinn, Anticevic, and Murray, 2020), which might have increased the similarities between change matrices and partitions.”

11. Modularity analysis – it appears the authors used a standard implementation of the Louvain algorithm. The default definition of Q defines a null expectation for within-cluster connectivity that is not appropriate for correlation matrices. With such data, the mean correlation can be appropriate.

We regret not describing our methods more clearly to avoid misunderstandings. We used an undirected weighted connection matrix with positive and negative correlations values (de-meaned) and the negative weights were treated asymmetrically (Rubinov and Sporns, 2011). This has now been clarified in the manuscript (in addition to a corrected reporting of the versatility curve, which was, erroneously, stemming from a non-de-meaned matrix in the original manuscript) (Materials and methods, section Experimental Design and Statistical Analysis):

“To identify clusters of correlations that could be compared across matrices, the community structure or modules in the matrices were obtained using the Louvain algorithm (V.D. Blondel, J-L., R., and E., 2008), part of the Brain Connectivity Toolbox (http://www.brain-connectivity-toolbox.net (Rubinov and Sporns, 2010)). […] To account for global brain changes, between-regional correlations were de-meaned before they were entered into the clustering analyses.”

And a further description is given in the section on Cluster stability analyses:

“As different clustering approaches often yield different results, we ran a series of post hoc analyses to confirm the validity of the cluster solution. […] Specifically, the 5-cluster solution resulted 7 times compared with once for the 3- and 8-cluster solutions, and twice for the 6- and 7-cluster solutions. Hence, this analysis supported the stability of the initial solution.”*12. Modularity analysis – the evaluation of different levels of γ is commendable. It is unclear why this is left to the end of the Results. This should either be placed at the beginning, as a way of justifying the 5-cluster solution, or the beginning should indicate what value of γ was used initially and contain a reference to subsequent analyses that investigate the issue of γ in more detail.*

We agree with the reviewers that this information should be presented earlier in the manuscript. In the re-organized Results section, this now follows just after the presentation of the cluster solution.

13. The similarity between the clusters and developmental origins is qualitative. Given that this is a core aim of the analysis, perhaps the authors could perform some inference by shuffling the labels and using a measure of partition similarity such as their normalized mutual information?

We understand the reviewer’s point, and agree that it could be good to have some kind of quantitative measure of overlap. We have discussed different ways of doing this, but have not come up with anything we believe is sufficiently solid. There may be approaches we are not aware of, but have kept the qualitative interpretations in the manuscript for now. Instead we acknowledge this directly in the Discussion:

“It must be noted as a limitation that the coherence between the developmental clusters and embryonic brain development is based on a qualitative judgement.”

14. Please explain why the cortex is treated as a single structure? Cortical development is regionally heterogeneous.

We certainly agree that cortical change is heterogeneous. The reason we did not include different cortical regions is that we assume that despite heterogeneity, within-cortical change is much more tightly integrated than cortical-subcortical change. Hence, including multiple cortical regions among the subcortical regions would likely have yielded an isolated cluster containing all cortical labels. We recently published a paper studying vertex-wise coordinated change of the cerebral cortex, please see: Continuity and Discontinuity in Human Cortical Development and Change From Embryonic Stages to Old Age – PubMed (nih.gov). This study is cited in the manuscript.

15. Sex is known to modify trajectory shape for cortical and subcortical structures. It would be good to show that the observed clustering holds for male and female subsamples. This would provide an important sensitivity analysis and also potential evidence for a sort of split half reliability.

We agree with the reviewer that multiple previous publications have focused on sex differences in brain development and aging. Running separate analyses for females and males would reduce power to approximately 50% and likely yield less stable solutions. In our experience, sex has minor, negligible effects on lifespan trajectories of subcortical volumes (see e.g. Minute effects of sex on the aging brain: a multisample magnetic resonance imaging study of healthy aging and Alzheimer's disease – PubMed (nih.gov)). Since we do not expect the major organizational principles of subcortical change to differ as a function of sex, we wish to report analyses for the full sample only. Still, we have included sex as a covariate in all analyses to account for possible confounding effects of sex.

16. It would also be important to show how ROIs cluster when you use the first derivatives from the gam fits in Figure 1. That would provide a complimentary approach to the inter-individual change method on which current work is built, and also help unpack some of the potential concerns re "fluid vs. tissue" and "white right vs. other" as two big potential drivers for findings. I would add plots for the ventricular components in Figure 1 too, as well as dendrograms for the primary heat map clustering used to order other matrices. My guess would be split one is CSF vs. tissue, and split 2 is white/white-rich tissue vs. other tissue.

The first derivatives would be based on group analyses, which we believe will address a potentially different question. It is conceivable that two regions that show very similar trajectories on a group level, and hence have similar derivatives, still show low within-subject change-change correlations. Although we agree that these can be interesting analyses, we believe they will not contribute to address the same questions as the original analyses in the manuscript. In light of the number of cluster analyses added to the revised manuscript, we are hesitant to further add to the complexity of the result. Regarding the question of contributions from WM and CSF, this is quite thoroughly treated in the revised manuscript, please see responses above.

17. In estimating APCs – how did the authors deal with were people with more than 2 scans, and the potential for including mid-scan age as a factor before correlating given the non-linear volume changes in development?

We apologize that this was not clearly explained in the manuscript. We have now added the following to the explanation of APCs were calculated (Results): “*If more than two time points were available, the first and the last were used to calculate APC.*” The reviewer is right that non-linear trajectories characterize brain development. Unfortunately, with relatively short follow up intervals (on average 1.7 years in our developmental sample), it is impossible to model non-linear changes, even in the very few cases where three timepoints were available.

18. I may have missed it but I did not expect the analysis of cognitive function. What purpose it is serving and does it really integrate in the theory of neurodevelopment? In addition, only one association seems significant after multiple testing and the difference seems to be localised in the 25-60 years old group for which you have the fewest observations (from Figure 1). Could this be due to a handful of outliers? Could you also add precision about which test was used? Maybe reporting several curves for difference quantiles of the distribution (instead of above/below average) would help visualise the effect?

We see the reviewers’ point. In response to this comment, we first decided to remove this section from the manuscript altogether, as it already contains a large number of comprehensive analyses. However, after further discussions among the co-authors, we came to the conclusion that it would be bad practice to remove these results at this stage, as the analyses were already conducted. Thus, we have removed the associated figure and kept the following shortened description in the main text (Results):

“Auxiliary analyses were done relating the clusters to general cognitive function (GCA) as measured by the Wechsler’s Abbreviated Scale of Intelligence (Wechsler, 1999) in the full LCBC sample, using sex and age as covariates. […] Importantly, only for Cluster 1 was a significant interaction between GCA and age found (F = 4.59, p = .01), suggesting that for the remaining clusters, age-trajectories did not differ significantly as a function of GCA (all p’s >.46).”

19. Figures 4 and 5 need a color scale

Yes, a color scale should have been included, thank you for spotting this. The previous Figure 4 is now part of Figure 1, which includes the color scale. A color scale has been added to Figure 3 (originally Figure 5).

20. Figure 2 is the main figure of result and I strongly suggest you expand the caption to improve self-readability. For example, the fact that the first row are results obtained on the LCBC sample. Clarify that the clustering was not recalculated for each matrix but is that of the first panel. In addition, the titles of the correlation plots could be clearer (e.g. clearly stating phenotypic and genetic correlations). Also the 4th panel is not a volumetric change-change relationship, which is the title of the figure.

We have now re-written the caption to the new Figures 1 and 2. The captions for the new figures read:

“Figure 1: Volumetric change-change relationships

Heat-maps represent pairwise correlations coefficients between volume change (annualized percent change) of the brain structures in development in the LCBC sample (left panel), aging in the LCBC sample (middle panel) and aging in the Lifebrain replication sample (right panels). The five clusters, delineated by the black lines, were derived from the developmental sample.”

“Figure 2 Genetic correlations

Left panel: Change-change correlations in development used to generate clusters. […] The five clusters, delineated by the black lines, were derived from the developmental sample.”

21. Abstract and introduction should announce that the focus is about volume of subcortical structures.

We agree that this should have been more clearly stated. In the Abstract, we now state:

“… we used graph theory to identify five clusters of coordinated development, indexed as patterns of correlated volumetric change in brain structures.”

In the Introduction, the following is included:

“Change was measures as annual percent change in the volume of a range of brain structures and areas. We hypothesized that volumetric changes in the developmental structures would tend to cluster according to embryonic principles, …”.

22. The discussion reads well but contains many statements that do not seem supported by the results or that sounded too definitive considering the analysis only focused on 16 volumetric measurements. It also lacks a limitation section.

We have gone through the discussion and modified the language. Specifically, we make sure that it is clear that we are discussing volumetric changes, and that statements do not exceed the empirical findings reported. We have also added a new section to the end of the Discussion: “Limitations: Caveats in interpreting brain changes from MRIs and further research”, cited above.

23. Some aspects of the text could be clearer. For example:Line 44 – it is unclear what the hypothesis is precisely.

We have now reformulated the sentence to make the hypothesis clearer (Abstract):

“We tested the hypothesis that genetically governed neurodevelopmental processes can be traced throughout life by assessing to which degree brain regions that develop together continue to change together through life.”

Line 59 – what does a topographic organization correspond to in this context? regionally specific?

Yes. To clarify, we have slightly revised the sentence:

“Cortical development follows a topographic organization through childhood and adolescence (Fjell et al., 2018; Krongold, Cooper, and Bray, 2017; Raznahan et al., 2011), meaning that regions of the cortex that can be distinguished from neighboring regions by different criteria such as structural and functional properties tend to develop together (see (Eickhoff, Constable, and Yeo, 2018) for a discussion of cortical topography in the context of neuroimaging).”

Line 61 – it is unclear what it means to follow the "genetic organization of the cortex".

We have reformulated the sentence to make this clearer:

“This topography is conserved through later development and aging (Fjell et al., 2018; Tamnes et al., 2013), closely following the genetic organization of the cortex, i.e. being controlled by overlapping sets of genes (Fjell et al., 2015).”

Line 79 – it is very hard to understand what this hypothesis refers to and what it predicts.

We agree that this was not clear. Hence, we have reformulated the description of this:

“On the other hand, a hypothesis is that genetically governed neurodevelopmental processes can be traced in the brain later in life (Chen et al., 2011; Satizabal et al., 2019). […] This has been shown for the comparably less plastic cortex (Fjell et al., 2015).”

Line 88 – it is unclear what a genetic anatomical architecture is.

To make this clear, we have re-formulated the sentence:

“Specifically, we tested how subcortical developmental volumetric change clustered across different structures, how similar this organization was in development versus aging, and whether clusters of change were influenced by shared genetics.”

Line 105 – please clarify: was the correlation across subjects?

Yes, exactly, we now state: “These APCs were correlated across participants between each pair of brain regions.”

Line 121 – it is unclear what "common factors" refers to here.

We have reformulated: “The extensive connectivity between cerebellum and cerebrum and similarities in development of WM in cerebellum and cerebrum may explain the latter finding.”

Line 217 – only a correlation rather than prediction has been shown.

We have reformulated: “…regional subcortical volumetric changes in aging follow a similar pattern as developmental changes in childhood, …”

Line 256 – please clarify: was consensus clustering run over 1000 runs at each γ?

We ran the consensus clustering 1000 times at the γ used in the main analysis, this is now clearly stated in the manuscript (see response above).

“Although the lifespan trajectories of subcortical structures are much more divergent than those for cortical regions.”Unclear to me how this is supported by results from the article, could you expand on how you conclude this?

This refers to previous literature, so we should have provided citations. We have slightly reformulated the sentence to make this clear: “Although the lifespan trajectories of subcortical structures have been shown to be much more divergent than those for cortical regions (Fjell et al., 2014; Walhovd et al., 2011), …”

“Mapping the developmental clusters to the adult part of the sample yielded highly different change trajectories.”Here it may help to cite figures and tables in Discussion – also is this the case for all clusters?

We agree, this now reads:

“Mapping the developmental clusters to the adult part of the sample yielded highly different change trajectories (see Figure 4). Except for clusters 3 and 4, which were characterized by mostly linear negative trajectories, differences in the shapes of the slopes were observed, suggesting that clusters identified in development continued to show independent trajectories of change through the rest of life.”

“Clusters 1, 3 and to a lesser extent 4 were related to general cognitive function in a mostly age-invariant manner, which implies that the relationship between cognitive function and subcortical volumes is established early in life.”This sounds like a huge overstatement. You studied only volume change of 16 regions – and the associations with cognition scores are not the most convincing.

We have now removed this section from the manuscript (see above).

“This means that regions that develop and change together through life tend to be governed by the same sets of genes.”Beyond my previous criticism, this reads as a rather large generality.

We have modified the statement: “This means that regions that develop and change together through life tend to be influenced by shared sets of genes.”

“…and are governed by distinct sets of genes.”What is this conclusion based on?

It is primarily based on the finding from the twin analyses that genetic change-change correlations show similar clustering to the developmental change-change correlations. We have modified the sentence somewhat in the revised manuscript:

“Subcortical change during childhood development can be organized in meaningful clusters, which are stable through life, tend to follow gradients of embryonic brain development and tend to be influenced by shared sets of genes.”

“It has also been argued that genes expressed in the subcortex generally are more region-specific and tend to evolve more rapidly than genes expressed in cortical regions”.I am not sure what you mean by "evolve more rapidly" is it that it is under greater selection pressure?

It means that they tend to change more during evolution, and thus that genes expressed in subcortical structures tend to be evolutionary more recent compared to genes expressed in the cortex. We have reformulated the text slightly for clarity:

“Although the subcortex is evolutionary older than the cortex, it has a higher proportion of evolutionarily more recent genes, and a higher evolutionary rate, which is a basic measure of evolution at the molecular level (Tuller et al., 2008). It has also been argued that genes expressed in the subcortex generally are more region-specific (Tuller et al., 2008; Zhang and Li, 2004).”

The SNP genetic correlation matrix was highly similar to the developmental change matrix, as demonstrated by the Mantel test (r = 0.57, p <.0005, see Figure 2).”I am not a big fan of qualitative judgments (e.g. highly). Especially that I assume r can vary between 0 and 1 so "highly" can be seen as an overstatement.As much as the Mantel test statistics are different from 0, are they also different from 1?

Agree! We have reformulated:

“The SNP genetic correlation matrix was more similar to the developmental change matrix than expected by chance, …”

We have done the same change to a second sentence also containing “highly similar”.

Generalized“Additive Mixed Models (GAMM)”Could you add some details about the maximal order of the splines you considered? How was the best model – best order selected?

Degree of smoothness was estimated as part of model fitting, and model selection was guided by minimizing AIC and BIC. This information was “hidden” in the table legend to Table 3. We have now moved it to the main text, stating: “Both Akaike Information Criterion (AIC) and Bayesian Information Criterion (BIC) were calculated to select among models and guard against over-fitting.”

24. “This revealed a close to perfect match between the adult genetic clusters and their embryonic origins.”You say grouping is extremely similar, but compared to the clusters presented in Figure 2, I feel that this is an overstatement. E.g. caudate, accumbens, putament, pallidum – two of those were in separate clusters previously. Also positive rG between cerebellum cortex and WM while this was negative before.

This specific statement referred to the match between the non-CSF clustering of the SNP-coheritability estimates from UKB and the main divisions of embryonic brain development. Thus, we expect certain differences from the plots presented in Figure 2. Still, we have moderated the statement:

“This revealed a match between the adult genetic clusters and their embryonic origins (Figure 3).”

25. “In all cases yielded the slope function the lowest IC values”. This is not true – see caudate and cortex.What to make from small AIC/BIC differences i.e. a marginally better fit? I wonder whether Table 3 is really adding anything, especially that the age trajectories are mostly descriptive?

You are right – thanks for spotting this! We have now corrected the statement. We agree with the reviewer that Table 3 is not critical to the manuscript. We think it is relevant to include it still, to yield some numbers to accompany Figure 5. However, we are open to remove it.

26. “We fitted the developmental trajectory of each cluster…”.I assume you took the total volume of each cluster? What is this analysis really adding? First it assumes the cluster is somewhat homogenous (see my second comment) and another problem is that the different volumes can have extremely different scales which makes interpreting the sum difficult.

Yes, we used the total volume of each. We agree with the reviewer that there certainly are differences within clusters, but as we show in the change-change matrix analyses, the differences within are smaller than the differences between the clusters. The individual differences between the clusters can be inspected in Figure 5, showing the age-trajectories of the individual structures. We understand the reviewers’ points, but still believe that showing the variable developmental and lifespan trajectories for these different clusters yield useful information about their differences. To make our view clearer, and to acknowledge the reviewers’ points, the text has been modified, so that it now reads:

“We fitted the developmental trajectory of each cluster by using the total volume of the structures within each cluster, […] Since the total volume was used, large structures will potentially influence the cluster trajectories more than smaller structures.”

27. “Using multivariate latent change score models, we calculated…”.Why did you not also use the annual symmetrized percent change you used previously? Especially that you want to compare rG to the phenotypic correlations you previously studied.

The main reason for this choice was that it was done in this way in the previous longitudinal subcortical twin study from ENIGMA (Bouwer et al.), of which the VETSA data used in the present study was drawn.

28. rGs from twin models are corrected for ICV, which does not seem to be the case for the GREML approach. Is this simply missing in the text?“…pair of the 16 brain sub-cortical structures, including the first ten principal components, sex and age as covariates.”

Sorry, this was an omission in the methods description. We have now corrected this:

“For the UKB SNP analyses, the volume measures of the 16 sub-cortical structures were corrected for ICV and, …”

29. “log likelihood test”. I am more used to it being referred to as the likelihood ratio test.

We have replaced “log likelihood test” with “likelihood ratio test”.

30. “restricted maximum likelihood methods” – (very) minor detail but REML is the optimisation approach used to estimate the parameters of what is a bivariate mixed model. A compromise may be to talk about GREML, which is not more correct but quite commonly used to refer to the LMM implemented in GCTA. For an analogy, it is as if you referred to the GAMM as a Restricted Marginal Likelihood method.

We have slightly reformulated the sentence to make it more accurate: “We used the bivariate linear mixed model with genome-based restricted maximum likelihood methods implemented in the program GCTA”.

31. “…due to heuristics in the algorithm…”. What are the heuristics here, is it random starting values?

Yes, the algorithm loops over all nodes in an order that is random for each run, and we now clarify this point (Methods, section Experimental Design and Statistical Analysis):

“The community structure may vary from run to run due to heuristics in the algorithm pertaining to the order in which the nodes are considered, […]”.

32. “Also, all reconstructed surfaces were inspected, and discarded if they did not pass internal quality control.” Surely, this does not apply to the UKB analyses. Did you perform any QC on them beyond the UKB provided screening?

No, UKB scans have already undergone some QC. To make this clear, we have added the following sentence (Methods, section on MRI data acquisition and analysis):

“UKB scans were quality checked by the UKB imaging team.”

33. “In addition, we removed participants suggested to be removed for genetic analysis by the UK Biobank team.” Could you add a precision about why this exclusion is recommended?

We have now updated the text with more detailed information:

“Ninety-one of these 481 participants had abnormal heterozygosity values, and the remaining were flagged out as outliers in heterozegosity/missing rate from the current QC files (ukb_sqc_VZ.csv) provided the most recent UK Biobank team.”.